# A Fourier Approach to Mixture Learning

**Mingda Qiao**[*]
Stanford University
mqiao@stanford.edu

**Guru Guruganesh**
Google Research
gurug@google.com

**Ankit Singh Rawat**
Google Research
ankitsrawat@google.com

**Avinava Dubey**
Google Research
avinavadubey@google.com

**Manzil Zaheer**
Google DeepMind
manzilzaheer@google.com

## Abstract

We revisit the problem of learning mixtures of spherical Gaussians. Given samples from mixture $\frac{1}{k} \sum_{j=1}^{k} \mathcal{N}(\mu_j, I_d)$, the goal is to estimate the means $\mu_1, \mu_2, \ldots, \mu_k \in \mathbb{R}^d$ up to a small error. The hardness of this learning problem can be measured by the *separation* $\Delta$ defined as the minimum distance between all pairs of means. Regev and Vijayaraghavan [2017] showed that with $\Delta = \Omega(\sqrt{\log k})$ separation, the means can be learned using $\mathrm{poly}(k, d)$ samples, whereas super-polynomially many samples are required if $\Delta = o(\sqrt{\log k})$ and $d = \Omega(\log k)$. This leaves open the low-dimensional regime where $d = o(\log k)$.

In this work, we give an algorithm that efficiently learns the means in $d = O(\log k / \log \log k)$ dimensions under separation $d/\sqrt{\log k}$ (modulo doubly logarithmic factors). This separation is strictly smaller than $\sqrt{\log k}$, and is also shown to be necessary. Along with the results of Regev and Vijayaraghavan [2017], our work almost pins down the critical separation threshold at which efficient parameter learning becomes possible for spherical Gaussian mixtures. More generally, our algorithm runs in time $\mathrm{poly}(k) \cdot f(d, \Delta, \epsilon)$, and is thus fixed-parameter tractable in parameters $d$, $\Delta$ and $\epsilon$.

Our approach is based on estimating the Fourier transform of the mixture at carefully chosen frequencies, and both the algorithm and its analysis are simple and elementary. Our positive results can be easily extended to learning mixtures of non-Gaussian distributions, under a mild condition on the Fourier spectrum of the distribution.

## 1 Introduction

Gaussian mixture models (GMMs) are one of the most well studied models for a population with different components. A GMM defines a distribution over the $d$-dimensional Euclidean space as the weighted sum of normal distributions $\sum_{i=1}^{k} w_i \cdot \mathcal{N}(\mu_i, \Sigma_i)$, which are specified by following quantities: the number of components $k \in \mathbb{N}$, the component means $\mu_i \in \mathbb{R}^d$, the component covariances $\Sigma_i \in \mathbb{R}^{d \times d}$, which are positive definite matrices, and the weights $w_i \geq 0$ that sum up to 1. In this work, we consider the uniform spherical case, where the weights $w_i$ are uniform $w_i = \frac{1}{k}$ and the covariance matrix $\Sigma_i = I_d$ is the identity matrix. The central problem in this setup is to *efficiently* estimate the means $\mu_1, \ldots, \mu_k$. To avoid degenerate cases such as when some of the means are the same, it is common to parameterize the problem by the separation of the means $\Delta$, which guarantees that $\|\mu_i - \mu_j\|_2 \geq \Delta$ for all $i \neq j$.

---

[*]Part of this work was done while working as an intern at Google Research.

36th Conference on Neural Information Processing Systems (NeurIPS 2022).

More precisely, the problem is to estimate the means $\mu_1, \ldots, \mu_k \in \mathbb{R}^d$ up to an error $\epsilon$ with runtime that is $\text{poly}(k, \frac{1}{\epsilon}, d)$ with as small a separation $\Delta$ as possible. There has been a long line of work on this problem which we survey in Section 1.3.

Recently, Regev and Vijayaraghavan [2017] showed that a separation $\Delta = \Omega(\sqrt{\log k})$ is strictly necessary when the dimension $d = \Omega(\log k)$. Two natural questions arise immediately. First, if $\Delta = \sqrt{\log k}$ is sufficient when $d = \Omega(\log k)$. Although the original work of Regev and Vijayaraghavan [2017] showed that it was information theoretically possible, an actual efficient algorithm was only recently developed by Liu and Li [2022] (who show nearly tight results). The second main question is determining the optimal separation in low dimensions when $d = o(\log k)$. Previously, even in $O(1)$ dimensions, the exact separation necessary was unknown. In this paper, we settle the second question and give nearly optimal bounds on the separation necessary in low dimensions (see Figure 1 for more details).

## 1.1 Overview of Results

We begin with a few definitions. A point set $\{x_1, x_2, \ldots, x_k\}$ is called $\Delta$-*separated* if $\|x_j - x_{j'}\|_2 \geq \Delta$ for any $j \neq j'$. We say that a Gaussian mixture is $\Delta$-separated (or has separation $\Delta$) if the means of its components are $\Delta$-separated. Two point sets $\{u_1, \ldots, u_k\}$ and $\{v_1, \ldots, v_k\}$ are $\epsilon$-*close* if for some permutation $\sigma$ over $[k]$, $\|u_j - v_{\sigma(j)}\|_2 \leq \epsilon$ holds for every $j$.

Our main result is an algorithm that efficiently learns the parameters of a mixture of $k$ spherical Gaussians under separation $\Delta \approx \frac{d}{\sqrt{\log k}} \cdot \text{poly}(\log \log k)$ in $d = O(\log k / \log \log k)$ dimensions. In the low-dimensional regime, this separation is strictly smaller than $\min\{\sqrt{\log k}, \sqrt{d}\}$, the smallest separation under which previous algorithms could provably learn the parameters in $\text{poly}(k)$ time.

**Theorem 1.1** (Upper bound, informal). *Let $P$ be a uniform mixture of $k$ spherical Gaussians in $d = O\left(\frac{\log k}{\log \log k}\right)$ dimensions with separation $\Delta = \Omega\left(\frac{d\sqrt{\log((\log k)/d)}}{\sqrt{\log k}}\right)$. There is a $\text{poly}(k)$-time algorithm that, given samples from $P$, outputs $k$ vectors that are w.h.p. $\epsilon$-close to the true means for $\epsilon = O(\Delta/\sqrt{d})$.*

See Theorem 2.1 and Remark 2.2 for a more formal statement of our algorithmic result, which holds for a wider range of separation $\Delta$ and accuracy parameter $\epsilon$. Our learning algorithm is provably correct for arbitrarily small $\Delta, \epsilon > 0$ (possibly with a longer runtime), whereas for most of the previous algorithms, there is a breakdown point $\Delta^*$ such that the algorithm is not known to work when $\Delta < \Delta^*$. Two exceptions are the algorithms of Moitra and Valiant [2010], Belkin and Sinha [2010], both of which allow an arbitrarily small separation but run in $e^{\Omega(k)}$ time. We also remark that the runtime of our algorithm scales as $\tilde{O}(k^3) \cdot f(d, \Delta, \epsilon)$, and is thus fixed-parameter tractable in parameters $d$, $\Delta$ and $\epsilon$.[2]

We complement Theorem 1.1 with an almost-matching lower bound, showing that the $d/\sqrt{\log k}$ separation is necessary for efficient parameter learning in low dimensions.

**Theorem 1.2** (Lower bound, informal). *For $d = O\left(\frac{\log k}{\log \log k}\right)$ and $\Delta = o\left(\frac{d}{\sqrt{\log k}}\right)$, there are two mixtures of $k$ spherical Gaussians in $\mathbb{R}^d$ such that: (1) both have separation $\Delta$; (2) their means are not $(\Delta/2)$-close; and (3) the total variation (TV) distance between them is $k^{-\omega(1)}$.*

See Theorem E.1 for a more formal version of the lower bound.

Theorem 1.1 and Theorem 1.2 together nearly settle the polynomial learnability of spherical Gaussian mixtures in the low-dimensional regime. Up to a doubly logarithmic factor, the "critical separation" where efficient learning becomes possible is $d/\sqrt{\log k}$. To the best of our knowledge, this was previously unknown even for $d = O(1)$.[3]

See Figure 1 below for a plot of our results in the context of prior work. The green regions cover the parameters $(\Delta, d)$ such that mixtures of $k$ spherical Gaussians in $d$ dimensions with separation $\Delta$ are

---

[2]While we focus on the uniform-weight case for simplicity, Theorem 1.1 can be easily extended to the setting where each weight is in $[1/(Ck), C/k]$ for some constant $C > 1$.

[3]An exception is the $d = 1$ case: A result of Moitra [2015] implies that $\Delta = \Omega(1/\sqrt{\log k})$ suffices (see Section 1.3), and a matching lower bound was given by Moitra and Valiant [2010].

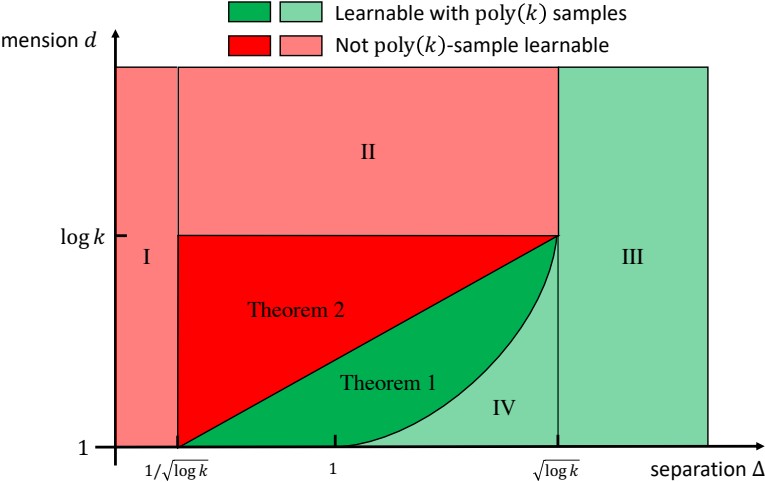

Figure 1: Region I is a direct corollary of Moitra and Valiant [2010, Proposition 15]. Regions II, III, and IV are shown by Regev and Vijayaraghavan [2017, Theorems 1.2, 1.3, and 1.4] respectively. The upper boundary of Region IV is the curve $\Delta = \sqrt{d}$. Theorems 1.1 and 1.2 settle the learnability in the remaining area, by proving that the line $\Delta = d/\sqrt{\log k}$ is the boundary between polynomial and super-polynomial sample complexities (up to a doubly-logarithmic factor).

learnable (up to $O(\Delta)$ error) using $\mathrm{poly}(k)$ samples.[4] The red regions contain the parameters under which polynomial-sample learning is provably impossible.

The algorithm that underlies Theorem 1.1 can be easily extended beyond the spherical Gaussian case. The following more general result states that for any distribution $\mathcal{D}$ whose the Fourier transform does not decay too fast, we can efficiently learn the parameters of a mixture of $k$ translated copies of $\mathcal{D}$. In the following, let $\mathcal{D}_\mu$ denote the distribution of $X + \mu$ when $X$ is drawn from $\mathcal{D}$.

**Theorem 1.3** (Learning more general mixtures, informal). *Let $P = \frac{1}{k}\sum_{j=1}^{k}\mathcal{D}_{\mu_j}$ for $\Delta$-separated $\mu_1, \ldots, \mu_k \in \mathbb{R}^d$. There is an algorithm that, given $\epsilon > 0$ and samples from $P$, runs in time*

$$\mathrm{poly}\left(k, \Delta/\epsilon, 1/\delta, \max_{\|\xi\|_2 \le M}\left|\mathbb{E}_{X \sim \mathcal{D}}\left[e^{i\xi^\top X}\right]\right|^{-1}\right)$$

*for some $\delta = \delta(\mathcal{D}, \epsilon)$ and $M = M(k, d, \Delta, \epsilon)$, and outputs $\hat{\mu}_1, \ldots, \hat{\mu}_k$ that are w.h.p. $\epsilon$-close to the true parameters.*

See Theorem F.1 and Corollary F.2 for a more formal statement of the runtime. Theorem 1.3 applies to many well-known distribution families that are specified by either a "location parameter" or a "scale parameter". Table 1 gives a few examples of applying Theorem 1.3 to mixtures of single-parameter univariate distributions; see Appendix F.2 and Appendix F.3 for more details.

**Limitations of our work.** The main limitation is that the positive results only apply to the regime that the dimension $d$ is logarithmic in the number of clusters, and that all clusters are translated copies of the same distribution. A concrete future direction would be to extend our results to learning mixtures of general Gaussians, even in one dimension.

## 1.2 Proof Overview

For simplicity, we focus on the one dimensional case that $P = \frac{1}{k}\sum_{j=1}^{k}\mathcal{N}(\mu_j, 1)$ for $\Delta$-separated means $\mu_1, \ldots, \mu_k \in \mathbb{R}$. We index the components such that $|\mu_1| \le |\mu_2| \le \cdots \le |\mu_k|$, and focus

---

[4]In terms of the computational complexity, all the green regions (except a small portion of Region III) admit efficient algorithms. The algorithm of Liu and Li [2022] is efficient when $\Delta = \Omega((\log k)^{1/2+c})$ for any $c > 0$ and thus almost covers Region III. For Region IV, Regev and Vijayaraghavan [2017] gave an efficient algorithm only for $d = O(1)$, whereas Theorem 1.1 covers the entire Region IV.

| Distribution | Parameter | Density Function | Runtime |
|:---:|:---:|:---:|:---:|
| Cauchy | $\mu$ | $\frac{1}{\pi(1+(x-\mu)^2)}$ | $O(k^3) \cdot e^{O(\sqrt{\log k}/\Delta)}$ |
| Logistic | $\mu$ | $\frac{e^{-(x-\mu)}}{(1+e^{-(x-\mu)})^2}$ | $O(k^3) \cdot e^{O(\sqrt{\log k}/\Delta)}$ |
| Laplace | $\mu$ | $\frac{1}{2}e^{-|x-\mu|}$ | $\tilde{O}(k^3/\Delta^5)$ |
| Exponential | $\ln \lambda$ | $\lambda e^{-\lambda x} \cdot \mathbb{1}\left[x \geq 0\right]$ | $O(k^3) \cdot e^{O(\sqrt{\log k}/\Delta)}$ |

Table 1: Implication of Theorem 1.3 for learning various families of mixtures of univariate distributions beyond Gaussians, assuming that the parameters of different components are $\Delta$-separated. The algorithm outputs $k$ parameters that are $O(\Delta)$-close to the true parameters.

on the following testing problem: Given $\epsilon > 0$ and samples from $P$, determine whether $\mu_1 = 0$ or $\mu_1 \geq \epsilon$, assuming that one of them is true. Note that this testing problem is not harder than estimating $\mu_1, \ldots, \mu_k$ up to error $\epsilon/3$—in the former case that $\mu_1 = 0$, one of the mean estimates would fall into $[-\epsilon/3, \epsilon/3]$, whereas all of them must be outside $(-2\epsilon/3, 2\epsilon/3)$ in the latter case. Conversely, as we will prove in Section 2, an algorithm for the testing version can be used for recovering the means as well.

**Examine the Fourier spectrum.** We start by examining the Fourier transform of $P$, $(\mathcal{F}P)(\xi) :=$ $\mathbb{E}_{X \sim P}\left[e^{i\xi X}\right]$, more commonly known in the literature as the *characteristic function*. Since the Fourier transform of a Gaussian is still a Gaussian, and a translation in the time domain shifts the phase in the frequency domain, we have

$$\mathbb{E}_{X \sim P}\left[e^{i\xi X}\right] = \frac{1}{k}\sum_{j=1}^{k} \mathbb{E}_{X \sim \mathcal{N}(\mu_j, 1)}\left[e^{i\xi X}\right] = \frac{e^{-\xi^2/2}}{k}\sum_{j=1}^{k} e^{i\mu_j \xi}. \tag{1}$$

We will focus on the quantity $A_\mu(\xi) := \sum_{j=1}^{k} e^{i\mu_j \xi}$, which is the "total phase" over the $k$ components of $P$. Equation (1) essentially states that $A_\mu(\xi)$ can be estimated by averaging $e^{i\xi X}$ over samples $X \sim P$.

The key observation is that each term $e^{i\mu_j \xi}$ of $A_\mu(\xi)$ behaves quite differently depending on the magnitude of $\mu_j$: If $\mu_j = 0$, $e^{i\mu_j \xi} = 1$ is a constant, whereas $e^{i\mu_j \xi}$ is a high-frequency wave when $|\mu_j|$ is large. This suggests that the two cases ($\mu_1 = 0$ and $|\mu_1| \geq \epsilon$) can be distinguished by estimating $A_\mu(\xi)$ at different frequencies. The cost of estimating $A_\mu(\xi)$, however, depends heavily on the frequency $\xi$ – Equation (1) together with a simple concentration bound shows that $O(k^2) \cdot e^{O(\xi^2)}$ samples are sufficient for estimating $A_\mu(\xi)$ up to a constant error.

Therefore, the crux of this approach is to find the minimum $M > 0$ such that the two cases can be distinguished by estimating $A_\mu(\xi)$ over $\xi \in [-M, M]$. (This is known as the *super-resolution* problem, which we discuss in Section 1.3.) The sample complexity of the testing problem can then be roughly bounded by $O(k^2) \cdot e^{O(M^2)}$. In the following, we explore different ways of picking $\xi$ from the range $[-M, M]$.

**Choosing $\xi$ randomly.** Our overall strategy is to draw $\xi$ randomly from some distribution $\mathcal{D}_\xi$ over interval $[-M, M]$ and evaluate $\mathbb{E}_{\xi \sim \mathcal{D}_\xi}[A_\mu(\xi)]$. We will argue that this expectation is very close to its first term $\mathbb{E}_{\xi \sim \mathcal{D}_\xi}\left[e^{i\mu_1 \xi}\right]$, which takes different values depending on whether $\mu_1 = 0$ or $|\mu_1| \geq \epsilon$. Then, $\mathcal{D}_\xi$ needs to be chosen such that: (1) There is a gap in the value of $\mathbb{E}_{\xi \sim \mathcal{D}_\xi}\left[e^{i\mu_1 \xi}\right]$ between the two cases; (2) $\left|\sum_{j=2}^{k} \mathbb{E}_\xi\left[e^{i\mu_j \xi}\right]\right|$ is small enough for the gap in the first term to be easily identified.

As a warmup, let $\mathcal{D}_\xi$ be the uniform distribution over $[0, M]$. A simple calculation shows that the gap in the first term $\mathbb{E}_{\xi \sim \mathcal{D}_\xi}\left[e^{i\mu_1 \xi}\right]$ between the two cases ($\mu_1 = 0$ or $|\mu_1| \geq \epsilon$) is lower bounded by $\Omega(\min\{M\epsilon, 1\})$, whereas the contribution from the $j$-th component satisfies $\left|\mathbb{E}_{\xi \sim \mathcal{D}_\xi}\left[e^{i\mu_j \xi}\right]\right| =$

$O\left(\frac{1}{M|\mu_j|}\right)$. Furthermore, the $\Delta$-separation between the means implies $|\mu_j| = \Omega(\Delta j)$. Thus,

$$\left|\sum_{j=2}^{k} \mathbb{E}_{\xi}\left[e^{i\mu_j\xi}\right]\right| \lesssim \sum_{j=2}^{k} \frac{1}{M|\mu_j|} \lesssim \frac{1}{M\Delta} \sum_{j=2}^{k} \frac{1}{j} \lesssim \frac{\log k}{M\Delta}.$$

The above is much smaller than $\min\{M\epsilon, 1\}$ if we set $M \gtrsim \max\left\{\frac{\log k}{\Delta}, \sqrt{\frac{\log k}{\Delta\epsilon}}\right\}$. Unfortunately, even when $\Delta$ and $\epsilon$ are constants, we have $O(k^2) \cdot e^{O(M^2)} = k^{O(\log k)}$ and the resulting sample complexity is already super-polynomial in $k$.

**A better choice of $\mathcal{D}_\xi$.** It turns out that choosing $\mathcal{D}_\xi$ to be a truncated Gaussian leads to a much lower sample complexity. For some $\sigma \ll M$, we draw $\xi \sim \mathcal{N}(0, \sigma^2)$ and then truncate it to $[-M, M]$. Without the truncation, the expectation of $e^{i\mu_j\xi}$ has a nice closed form:

$$\mathbb{E}_{\xi\sim\mathcal{N}(0,\sigma^2)}\left[e^{i\mu_j\xi}\right] = e^{-\sigma^2\mu_j^2/2},$$

which is exactly the Fourier weight of $\mathcal{N}(0, \sigma^2)$ at frequency $\mu_j$. Note that this decreases very fast as $|\mu_j|$ grows, compared to the previous rate of $\frac{1}{M|\mu_j|}$ when $\mathcal{D}_\xi$ is uniform.

It again follows from a simple calculation that: (1) Depending on whether $\mu_1 = 0$ or $|\mu_1| \geq \epsilon$, the gap between $\mathbb{E}_{\xi}\left[e^{i\mu_1\xi}\right]$ is $\Omega(\min\{\sigma^2\epsilon^2, 1\})$; (2) The total contribution from $j = 2, 3, \ldots, k$ is upper bounded by

$$\left|\sum_{j=2}^{k} \mathbb{E}_{\xi\sim\mathcal{N}(0,\sigma^2)}\left[e^{i\mu_j\xi}\right]\right| = \sum_{j=2}^{k} e^{-\sigma^2\mu_j^2/2} \leq \sum_{j=2}^{k} e^{-\Omega(\sigma^2\Delta^2 j^2)} = e^{-\Omega(\sigma^2\Delta^2)},$$

where the second step applies $|\mu_j| = \Omega(\Delta j)$ and the last step holds assuming $\sigma = \Omega(1/\Delta)$. In addition, we need to deal with the error incurred by the truncation. The Gaussian tail bounds imply that $|\xi| \geq M$ happens with probability $e^{-\Omega(M^2/\sigma^2)}$. Since $|A_\mu(\xi)| \leq k$ for any $\xi$, the noise from the truncation is at most $k \cdot e^{-\Omega(M^2/\sigma^2)}$ in magnitude.

It remains to choose $M, \sigma > 0$ that satisfy the two inequalities:

$$e^{-\Omega(\sigma^2\Delta^2)} \ll \min\{\sigma^2\epsilon^2, 1\} \quad \text{and} \quad k \cdot e^{-\Omega(M^2/\sigma^2)} \ll \min\{\sigma^2\epsilon^2, 1\}.$$

It suffices to choose $\sigma \lesssim \frac{1}{\Delta}\sqrt{\log\frac{\Delta}{\epsilon}}$ and $M \lesssim \frac{1}{\Delta}\sqrt{\log\frac{\Delta}{\epsilon}}\sqrt{\log\frac{k\Delta}{\epsilon}}$. For $\epsilon = \Omega(\Delta)$, the sample complexity $O(k^2) \cdot e^{O(M^2)}$ reduces to $k^{O(1/\Delta^2)}$, which is polynomial in $k$ for any fixed $\Delta$.

We note that in the above derivation, the assumption that each cluster of $P$ is a Gaussian is only applied in Equation (1) (through the Fourier transform). For any distribution $\mathcal{D}$ over $\mathbb{R}$ and $P = \frac{1}{k}\sum_{j=1}^{k}\mathcal{D}_{\mu_j}$, the quantity $A_\mu(\xi) = \sum_{j=1}^{k} e^{i\mu_j\xi}$ can still be read off from the Fourier transform of $P$ at frequency $\xi$, except that the extra factor $e^{-\xi^2/2}$ becomes $(\mathcal{F}\mathcal{D})(\xi)$, the Fourier transform of $\mathcal{D}$ at frequency $\xi$. This observation leads to our algorithm for learning a mixture of multiple translated copies of $\mathcal{D}$ (Theorem 1.3).

**Gaussian truncation.** For the spherical Gaussian case, however, the above only gives an algorithm with runtime $k^{O(1/\Delta^2)}$, which becomes super-polynomial as soon as $\Delta = o(1)$, falling short of achieving the near-optimal separation of $\sqrt{\frac{\log\log k}{\log k}}$ in Theorem 1.1.

We further improve our algorithm for the spherical Gaussian case using a "Gaussian truncation" technique. Intuitively, when deciding whether the mixture $P$ contains a cluster with mean zero, a sample $X \sim P$ is much more informative when $|X|$ is small. This motivates us to focus on samples with a small magnitude via a truncation.

We apply such a truncation in a soft way—weighting each sample $X$ with $e^{-X^2/2}$. This turns out to be sufficiently effective while keeping the entire analysis simple. Note that the weighting effectively multiplies the mixture $P$ with the standard Gaussian $\mathcal{N}(0, 1)$ pointwise. The result is

still a (un-normalized) mixture of Gaussians: Up to a constant factor, the pointwise product is $\frac{1}{k}\sum_{j=1}^{k} e^{-\mu_j^2/4} \cdot \mathcal{N}(\mu_j/2, 1/2)$. Consequently, if we repeat the analysis from previous paragraphs to this weighted mixture, the noise coming from components with large $|\mu_j|$ become even smaller. This eventually allows us to learn the mixture efficiently at separation $\Delta = \sqrt{\frac{\log\log k}{\log k}}$. We prove Theorem 1.1 via a natural extension of this analysis to the $d$-dimensional case.

**Lower bound.** Our proof of Theorem 1.2 follows an approach similar to those in the previous lower bound proofs of Moitra and Valiant [2010], Hardt and Price [2015], Regev and Vijayaraghavan [2017]: First, construct two sets of well-separated points $\{\mu_1^{(P)}, \ldots, \mu_k^{(P)}\}$ and $\{\mu_1^{(Q)}, \ldots, \mu_k^{(Q)}\}$ with (approximately) matching lower-order moments, i.e., $\frac{1}{k}\sum_{j=1}^{k}\left[\mu_j^{(P)}\right]^{\otimes t} \approx \frac{1}{k}\sum_{j=1}^{k}\left[\mu_j^{(Q)}\right]^{\otimes t}$ for every small $t$. Then, show that these matching moments imply that after a convolution with Gaussian, the resulting mixtures are close in TV-distance and thus hard to distinguish.

In more detail, similar to the proof of Regev and Vijayaraghavan [2017], we start by choosing $N$ arbitrary points from a small $\ell_2$ ball, such that they are $\Delta$-separated. The main difference is that Regev and Vijayaraghavan [2017] pick $N \gg k$, and show that among all the $\binom{N}{k}$ possible mixtures, there are at least two mixtures with similar moments via a pigeon-hole type argument. In contrast, we work in the $N \ll k$ regime, and obtain two point sets with matching lower-order moments by slightly perturbing these $N$ points in opposite directions. The existence of such a good perturbation is shown via a careful application of the Borsuk–Ulam Theorem, which is inspired by a similar application in Hardt and Price [2015].

## 1.3 Related Work

**Learning Gaussian mixture models.** Most closely related to this paper is the line of work in the theoretical computer science literature on algorithms that provably learn mixtures of Gaussians. The pioneering work of Dasgupta [1999] showed that an $\Omega(\sqrt{d})$ separation between the means of different components is sufficient for the samples to be easily clustered. Several subsequent work (e.g., Vempala and Wang [2004], Arora and Kannan [2005], Achlioptas and McSherry [2005], Kannan et al. [2005], Dasgupta and Schulman [2007], Brubaker and Vempala [2008]) further generalized this result and the separation condition is relaxed to $\Delta = \Omega((\min\{k,d\})^{1/4} \cdot \text{poly}(\log k, \log d))$ by Vempala and Wang [2004]. We refer the readers to Regev and Vijayaraghavan [2017] for a more detailed survey of these results.

Moitra and Valiant [2010] and Belkin and Sinha [2010] gave the first algorithms for learning general mixtures of Gaussians (with different unknown covariances) under an arbitrarily small separation between different components. Both algorithms are based on the method of moments, and run in time polynomial in $d$ and the inverse of the minimum TV-distance between different components when $k = O(1)$. For large $k$, however, the runtime is exponential in $k$.

Regev and Vijayaraghavan [2017] showed that $\text{poly}(k,d)$ samples are sufficient to estimate the means of a mixture of spherical Gaussians, if the means are separated by $\Delta = \Omega(\sqrt{\log k})$. Unfortunately, their algorithm involves an exhaustive search and is computationally inefficient. Subsequently, three concurrent work [Diakonikolas et al., 2018, Hopkins and Li, 2018, Kothari et al., 2018] developed algorithms based on the Sum-of-Squares hierarchy that run in time $(dk)^{\text{poly}(1/\gamma)}$ and learn mixtures with separation $\Delta = \Omega(k^\gamma)$. In particular, setting $\gamma \approx \frac{\log\log k}{\log k}$ gives a quasi-polynomial time algorithm that achieves the $\sqrt{\log k}$ separation in Regev and Vijayaraghavan [2017]. A very recent work of Liu and Li [2022] made further progress towards learning $\Omega(\sqrt{\log k})$-separated mixtures efficiently, by giving a polynomial-time algorithm that succeeds under separation $(\log k)^{1/2+c}$ for any constant $c > 0$.

**Lower bounds.** On the lower bound side, Moitra and Valiant [2010] first proved, by explicitly constructing a hard instance, that learning a mixture of $k$ Gaussians with separation $\Delta = \Omega(1/\sqrt{k})$ requires $e^{\Omega(k)}$ samples even in $d = 1$ dimension. Hardt and Price [2015] focused on the regime where $k = O(1)$ and the recovery error $\epsilon$ goes to zero, and showed that $\Omega(\epsilon^{2-6k})$ samples are necessary. Anderson et al. [2014] proved a lower bound that strengthens the result of Moitra and Valiant [2010]: Separation $\Delta = 1/\text{poly}(k)$ is insufficient for the mixture to be learnable with polynomially many

samples, even when the means of the Gaussians are drawn uniformly at random from $[0,1]^d$ for $d = O(\log k / \log \log k)$.

Regev and Vijayaraghavan [2017] proved that in $d = \Omega(\log k)$ dimensions, no polynomial-sample algorithm exists if $\Delta = o(\sqrt{\log k})$. In particular, even when the means are chosen randomly, the resulting mixture is still hard to learn with high probability. This lower bound, together with their positive result for the $\Delta = \Omega(\sqrt{\log k})$ regime, shows that $\sqrt{\log k}$ is the critical separation threshold in high dimensions.

If we restrict our attention to statistical query (SQ) algorithms, Diakonikolas et al. [2017] proved that any SQ algorithm needs $d^{\Omega(k)}$ queries if $d \geq \text{poly}(k)$ and non-spherical components are allowed.

**Density estimation.**    In contrast to the parameter estimation setting that we focus on, the *density estimation* setting requires the learning algorithm to output (the representation of) a hypothesis distribution $\hat{P}$ that is close to the mixture $P$ in TV-distance. Ashtiani et al. [2020] proved that the sample complexity of learning Gaussian mixtures up to a TV-distance of $\epsilon$ is $\Theta(d^2 k/\epsilon^2)$ in general, and $\tilde{\Theta}(dk/\epsilon^2)$ if all components are axis-aligned. Unfortunately, their learning algorithms are not computationally efficient and run in time exponential in $k$. For the spherical case, Diakonikolas and Kane [2020] gave a more efficient algorithm that runs in quasi-polynomial time $\text{poly}(d) \cdot (k/\epsilon)^{O(\log^2 k)}$. In the proper learning setting (i.e., $\hat{P}$ must be a Gaussian mixture), Suresh et al. [2014] gave an algorithm that takes $\text{poly}(dk/\epsilon)$ samples and runs in time $\text{poly}(d) \cdot (k/\epsilon)^{O(k^2)}$.

**Super-resolution, and mixture learning using Fourier analysis.**    Finally, we acknowledge that the Fourier approach that we explore in this work is fairly natural, and similar ideas have appeared in prior work. In particular, our approach is closely related to the super-resolution problem (Donoho [1992], Candès and Fernandez-Granda [2013, 2014])—to recover $k$ unknown locations $\mu_1, \ldots, \mu_k \in \mathbb{R}^d$ from (exact or noisy) observations of the form $\sum_{j=1}^{k} e^{i\mu_j^\top \xi}$ for $\|\xi\| \leq M$, where $M$ is called the cutoff frequency, and the norm is typically $\ell_2$ or $\ell_\infty$. In comparison, our approach (outlined in Section 1.2) differs in two aspects: (1) We focus on a simpler testing version of the problem—to decide whether one of the locations is near a given reference point; (2) The Gaussian truncation significantly down-weights the observation coming from the points that are far from the reference point.

For the $d = 1$ case, Moitra [2015] showed that a cutoff frequency of $M = O(1/\Delta)$ suffices if the points are $\Delta$-separated. This implies an algorithm that efficiently learns spherical Gaussian mixtures in one dimension under separation $1/\sqrt{\log k}$, which is (nearly) recovered by our positive result. For the general case, Huang and Kakade [2015] gave an algorithm that provably works for ($\ell_2$) cutoff frequency $M = O((\sqrt{d \log k} + \log k)/\Delta)$. When applied to learning GMMs, however, their algorithm requires separation $\sqrt{d}$. It is conceivable that the algorithm of Huang and Kakade [2015], equipped with appropriate modifications and a tighter analysis (e.g., the approach of Chen and Moitra [2021]), *might* give a guarantee similar to ours, but no such analysis is explicit in the literature to the best of our knowledge. Moreover, our approach leads to an arguably simpler algorithm with an elementary analysis.

More recently, Chakraborty and Narayanan [2020] gave an algorithm that is similar to ours for learning mixtures of spherical Gaussians via deconvolving the mixture. However, their algorithm is only shown to work when $\Delta = \Omega(\sqrt{d})$. Chen et al. [2020] studied the problem of learning mixtures of linear regressions (MLRs), which can be reduced to estimating the minimum variance in a mixture of zero-mean Gaussians. They solved this problem by estimating the *Fourier moments* – the moments of the Fourier transform, and gave the first sub-exponential time algorithm for learning MLRs. Chen and Moitra [2021] studied learning mixtures of Airy disks, a problem that is motivated by the physics of diffraction. Their algorithm also proceeds by first estimating the Fourier transform of the mixture, and then dividing it pointwise by the Fourier spectrum of the "base" distribution.

## 1.4  Organization of the Paper

In Section 2, we formally state our main algorithmic result as well as our main technical theorem, which addresses a testing version of the problem. We then sketch how our upper bound follows from

a simple reduction from parameter learning to testing. In Section 3, we give a simple algorithm that solves this testing version by examining the Fourier transform of the mixture.

The pseudocode of our algorithms are presented in Appendix A. We defer a few technical proofs to Appendices B through D. We prove our lower bound result (Theorem 1.2) in Appendix E. Finally, in Appendix F, we formally state and prove the guarantees for learning non-Gaussian mixtures and present a few applications of this result.

## 2 From Learning to Testing

We first state our main positive result – the formal version of Theorem 1.1.

**Theorem 2.1.** *Let $P = \frac{1}{k}\sum_{j=1}^{k}\mathcal{N}(\mu_j, I_d)$ be a uniform mixture of $k \geq 2$ spherical Gaussians with $\Delta$-separated means in $\mathbb{R}^d$ and $\epsilon < \min\{\Delta/100, \Delta/(32\sqrt{\min\{d, \ln k\}})\}$. There is an algorithm (Algorithm 1) that, given samples from $P$, outputs $k$ vectors that are $\epsilon$-close to $\mu_1, \dots, \mu_k$ with high probability. The runtime (and thus the sample complexity) of the algorithm is upper bounded by*

$$O((\Delta/\epsilon)^4 k^3 \log^2 k) \cdot \max\{(d/\epsilon)^{O(d)}, d^{O(d)}\} \cdot e^{O(M^2)},$$

*where $M^2 \lesssim \frac{1}{\Delta^2}(\min\{d, \log k\} + \log\frac{\Delta}{\epsilon}) \cdot (\min\{d + \log k, d\log(2 + \frac{d}{\Delta^2})\} + \log\frac{\Delta}{\epsilon})$.*

*Remark* 2.2. When $d = O\left(\frac{\log k}{\log\log k}\right)$ and $\Delta/\epsilon = \Theta(\sqrt{d})$, the runtime can be simplified into

$$\text{poly}(k) \cdot \exp\left(O\left(\frac{d}{\Delta^2}\min\left\{\log k, d\log\left(2 + \frac{d}{\Delta^2}\right)\right\}\right)\right),$$

which is $\text{poly}(k)$ if $\Delta = \Omega\left(\frac{d}{\sqrt{\log k}} \cdot \sqrt{\log\frac{\log k}{d}}\right)$. When $d = o(\log k)$, this condition is strictly looser than both $\Delta = \Omega(\sqrt{d})$ and $\Delta = \Omega(\sqrt{\log k})$.

We prove Theorem 2.1 by reducing the parameter learning problem into the following testing version: Given samples from mixture $P$, determine whether $P$ contains a cluster with a mean that is close to a given guess $\mu^* \in \mathbb{R}^d$. Formally, we prove the following theorem:

**Theorem 2.3.** *Let $P = \frac{1}{k}\sum_{j=1}^{k}\mathcal{N}(\mu_j, I_d)$ be a uniform mixture of $k \geq 2$ spherical Gaussians with $\Delta$-separated means in $\mathbb{R}^d$. Let $\epsilon < \min\{\Delta/100, \Delta/(32\sqrt{\min\{d, \ln k\}})\}$ and $\mu^* \in \mathbb{R}^d$. There is an algorithm (Algorithm 2) that, given samples from $P$, either "accepts" or "rejects", such that it:*

- *Accepts with probability $\geq 2/3$ if $\min_{j \in [k]} \|\mu_j - \mu^*\|_2 \leq \epsilon/2$.*
- *Rejects with probability $\geq 2/3$ if $\min_{j \in [k]} \|\mu_j - \mu^*\|_2 \geq \epsilon$.*

*The runtime (and thus the sample complexity) of the algorithm is upper bounded by $O(k^2(\Delta/\epsilon)^4) \cdot e^{O(d+M^2)}$, where $M^2 \lesssim \frac{1}{\Delta^2}(\min\{d, \log k\} + \log\frac{\Delta}{\epsilon}) \cdot (\min\{d + \log k, d\log(2 + \frac{d}{\Delta^2})\} + \log\frac{\Delta}{\epsilon})$.*

Assuming the above theorem, the proof of our main theorem is straightforward and is thus deferred to Appendix C. The proof proceeds by first drawing a few samples $X_1, \dots, X_N$ from the mixture, and then running the tester (from Theorem 2.3) to decide whether each $X_i$ is close to one of the mean vectors of the mixture. Then, a simple argument shows that the samples that the tester accepts can be easily clustered to recover the means. Note that if a sample $X_i$ comes from the $j$-th component $\mathcal{N}(\mu_j, I_d)$, there is a decent probability (e.g., $\Omega(\epsilon)$ when $d = 1$) that $X_i$ is $\epsilon$-close to $\mu_j$. Thus, we can guarantee that $X_1, \dots, X_N$ contain good guesses for all the $k$ means for moderately large $N$.

## 3 Solve the Testing Problem using Fourier Transform

Now we solve the testing problem and prove Theorem 2.3. We assume without loss of generality that $\mu^* = 0$, since we can reduce to this case by subtracting $\mu^*$ from each sample from $P$. We also re-index the unknown means $\mu_1, \dots, \mu_k$ so that $\|\mu_j\|_2$ is non-decreasing in $j$. The testing problem is then equivalent to deciding whether $\|\mu_1\|_2 \leq \epsilon/2$ or $\|\mu_1\|_2 \geq \epsilon$.

The following simple lemma, which we prove in Appendix D, gives the Fourier transform of the mixture $P$ when a "Gaussian truncation" of $e^{-\|x\|_2^2/2}$ is applied.

**Lemma 3.1.** *For $P = \frac{1}{k}\sum_{j=1}^{k} \mathcal{N}(\mu_j, I_d)$ and any $\xi \in \mathbb{R}^d$,*

$$\mathop{\mathbb{E}}_{X \sim P}\left[e^{-\|X\|_2^2/2} \cdot e^{i(\xi^\top X)}\right] = \frac{e^{-\|\xi\|_2^2/4}}{2^{d/2}k}\sum_{j=1}^{k} e^{-\|\mu_j\|_2^2/4} \cdot e^{i(\mu_j^\top \xi/2)} = \frac{e^{-\|\xi\|_2^2/4}}{2^{d/2}k} \cdot A_\mu(\xi),$$

*where we define $A_\mu(\xi) := \sum_{j=1}^{k} e^{-\|\mu_j\|_2^2/4} \cdot e^{i(\mu_j^\top \xi/2)}$.*

We will show that $A_\mu(\xi)$ behaves quite differently depending on whether $\|\mu_1\|_2 \leq \epsilon/2$ or $\|\mu_1\|_2 \geq \epsilon$. Thus, we can solve the testing problem by estimating $A_\mu(\xi)$ for carefully chosen $\xi$. Concretely, we will draw $\xi$ from $\mathcal{N}(0, \sigma^2 I_d)$ and then truncate it to $B(0, M) := \{x \in \mathbb{R}^d : \|x\|_2 \leq M\}$, for parameters $M, \sigma > 0$ to be chosen later. Formally, we focus on the following expectation:

$$T_\mu := \mathop{\mathbb{E}}_{\xi \sim \mathcal{N}(0, \sigma^2 I_d)}[A_\mu(\xi) \cdot \mathbb{1}[\|\xi\|_2 \leq M]].$$

The key step of our proof of Theorem 2.3 is to show that $T_\mu$ is close to $e^{-(\sigma^2/2+1)\|\mu_1\|_2^2/4}$, and is thus helpful for deciding whether $\|\mu_1\|_2 \leq \epsilon/2$ or $\|\mu_1\|_2 \geq \epsilon$. The following lemma, the proof of which is relegated to Appendix D, helps us to bound the difference between $T_\mu$ and $e^{-(\sigma^2/2+1)\|\mu_1\|_2^2/4}$.

**Lemma 3.2.** *For any $M, \sigma > 0$ that satisfy $M^2/\sigma^2 \geq 5d$,*

$$T_\mu = \sum_{j=1}^{k} e^{-(\sigma^2/2+1)\|\mu_j\|_2^2/4} + O\left(e^{-M^2/(5\sigma^2)}\right) \cdot \sum_{j=1}^{k} e^{-\|\mu_j\|_2^2/4},$$

*where the $O(x)$ notation hides a complex number with modulus $\leq x$.*

By Lemma 3.2, the difference $T_\mu - e^{-(\sigma^2/2+1)\|\mu_1\|_2^2/4}$ is given by

$$\sum_{j=2}^{k} e^{-(\sigma^2/2+1)\|\mu_j\|_2^2/4} + O\left(e^{-M^2/(5\sigma^2)}\right) \cdot \sum_{j=1}^{k} e^{-\|\mu_j\|_2^2/4}.$$

Let $S_1 := \sum_{j=2}^{k} e^{-(\sigma^2/2+1)\|\mu_j\|_2^2/4}$ and $S_2 := \sum_{j=1}^{k} e^{-\|\mu_j\|_2^2/4}$ denote the two summations above. We have the following upper bounds on $S_1$ and $S_2$, which we prove in Appendix D:

*Claim 3.3.* Assuming $(\sigma^2/2 + 1)\Delta^2 \geq 100\min\{\ln k, d\}$, $S_1 \leq 2e^{-(\sigma^2/2+1)\Delta^2/64} \cdot \min\{k, 2^d\}$.

*Claim 3.4.* We have $S_2 \leq \begin{cases} 2, & \Delta^2 \geq 100d, \\ 1 + \frac{267d}{\Delta^2} \cdot \max\left\{\left(\frac{32d}{\Delta^2}\right)^{d/2}, 1\right\}, & \Delta^2 < 100d. \end{cases}$ Furthermore, $S_2 \leq 10 \cdot \min\left\{k, 1 + \left(\frac{32d}{\Delta^2}\right)^{d/2+1}\right\}$.

Now we put all the pieces together and prove Theorem 2.3.

*Proof of Theorem 2.3.* By Lemma 3.1,

$$\begin{aligned}
T_\mu &= \mathop{\mathbb{E}}_{\xi \sim \mathcal{N}(0, \sigma^2 I_d)}[A_\mu(\xi) \cdot \mathbb{1}[\|\xi\|_2 \leq M]] \\
&= \mathop{\mathbb{E}}_{\substack{X \sim P \\ \xi \sim \mathcal{N}(0, \sigma^2 I_d)}}\left[2^{d/2}k \cdot e^{\|\xi\|_2^2/4} \cdot e^{-\|X\|_2^2/2} \cdot e^{i(\xi^\top X)} \cdot \mathbb{1}[\|\xi\|_2 \leq M]\right].
\end{aligned}$$

Since the term inside the expectation has modulus $\leq k \cdot e^{O(d+M^2)}$, a Chernoff bound implies that we can estimate $T_\mu$ up to any additive error $\gamma > 0$ using $O((k/\gamma)^2) \cdot e^{O(d+M^2)}$ samples from $P$.

By Lemma 3.2 and our definition of $S_1$ and $S_2$, assuming $M^2/\sigma^2 \geq 5d$,

$$\left|T_\mu - e^{-(\sigma^2/2+1)\|\mu_1\|_2^2/4}\right| \leq S_1 + e^{-M^2/(5\sigma^2)}S_2.$$

In the rest of the proof, we will pick $\sigma$ and $M$ carefully so that both $S_1$ and $e^{-M^2/(5\sigma^2)}S_2$ are upper bounded by $\gamma := (\sigma^2/2 + 1)\epsilon^2/64$. Assuming this, we are done: We can simply estimate $T_\mu$

up to an additive error of $\gamma$. Let $\widehat{T_\mu}$ denote the estimate. We accept if and only if $\operatorname{Re}\widehat{T_\mu} \geq \theta \coloneqq \frac{1}{2}\left[e^{-(\sigma^2/2+1)\epsilon^2/16} + e^{-(\sigma^2/2+1)\epsilon^2/4}\right]$. Indeed, suppose that $\|\mu_1\|_2 \leq \epsilon/2$, we have

$$\operatorname{Re}\widehat{T_\mu} \geq \operatorname{Re}T_\mu - \gamma \geq e^{-(\sigma^2/2+1)\|\mu_1\|_2^2/4} - (S_1 + e^{-M^2/(5\sigma^2)}S_2) - \gamma \geq e^{-(\sigma^2/2+1)\epsilon^2/16} - 3\gamma.$$

Similarly, $\operatorname{Re}\widehat{T_\mu} \leq e^{-(\sigma^2/2+1)\epsilon^2/4} + 3\gamma$ if $\|\mu_1\|_2 \geq \epsilon$. If we set $\sigma$ such that $(\sigma^2/2+1)\epsilon^2 \leq 1$, we have $e^{-(\sigma^2/2+1)\epsilon^2/16} - e^{-(\sigma^2/2+1)\epsilon^2/4} \geq \frac{1}{8}(\sigma^2/2+1)\epsilon^2 = 8\gamma$, which implies $\operatorname{Re}\widehat{T_\mu} > \theta$ in the former case and $\operatorname{Re}\widehat{T_\mu} < \theta$ in the latter case. Therefore, our algorithm decides correctly.

**Choice of parameters.** Claim 3.3 implies that if we set $\sigma^2/2 + 1 = \frac{512}{\Delta^2}\left(\min\{d, \ln k\} + \ln\frac{\Delta}{\epsilon}\right)$, we can ensure $S_1 \leq (\sigma^2/2+1)\epsilon^2/64 = \gamma$. Furthermore, this choice of $\sigma$ and the assumption $\epsilon < \min\{\Delta/100, \Delta/(32\sqrt{\min\{d, \ln k\}})\}$ guarantee the condition $(\sigma^2/2+1)\epsilon^2 \leq 1$ that we need.

It remains to pick $M$ such that $e^{-M^2/(5\sigma^2)} \cdot S_2 \leq \gamma = (\sigma^2/2+1)\epsilon^2/64$. We also need $M^2/\sigma^2 \geq 5d$ to ensure that Lemma 3.2 can be applied. It suffices to let $M^2 \geq 5\sigma^2 \cdot \left(d + \ln S_2 + \ln\frac{64}{(\sigma^2/2+1)\epsilon^2}\right)$. Our choice of $\sigma$ guarantees $\ln\frac{64}{(\sigma^2/2+1)\epsilon^2} \leq 2\ln\frac{\Delta}{\epsilon}$, so it is in turn sufficient to pick $M$ such that

$$M^2 = \frac{5120}{\Delta^2}\left(\min\{d, \ln k\} + \ln\frac{\Delta}{\epsilon}\right)\left(d + 2\ln\frac{\Delta}{\epsilon} + \ln S_2\right).$$

Applying Claim 3.4 shows that $M$ can be chosen such that

$$M^2 \lesssim \frac{1}{\Delta^2}\left(\min\{d, \log k\} + \log\frac{\Delta}{\epsilon}\right) \cdot \left(\min\left\{d + \log k, d\log\left(2 + \frac{d}{\Delta^2}\right)\right\} + \log\frac{\Delta}{\epsilon}\right).$$

**Runtime.** The runtime of our algorithm is dominated by the number of samples drawn from $P$—$O((k/\gamma)^2) \cdot e^{O(d+M^2)}$, where $\gamma = \Theta((\sigma^2+1)\epsilon^2)$. Plugging our choice of $\sigma$ into $\gamma$ gives $1/\gamma = O((\Delta/\epsilon)^2)$. The runtime can thus be upper bounded by $O(k^2(\Delta/\epsilon)^4) \cdot e^{O(d+M^2)}$. $\qquad\square$

## Acknowledgments

We would like to thank the anonymous reviewers of earlier versions of this paper for their suggestions on the presentation and for pointers to the literature.

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
