# A   Pseudocode of Algorithms

The pseudocode of our algorithms for learning and testing spherical Gaussian mixtures follow immediately from the proofs of Theorem 2.1 and Theorem 2.3. In the following, $P$ is a uniform mixture of $k$ spherical Gaussians in $\mathbb{R}^d$ with unknown $\Delta$-separated means.

---

**Algorithm 1:** Learning Gaussian Mixtures via Testing

---

**Input:** Sample access to mixture $P$. Accuracy parameter $\epsilon > 0$.
**Output:** Mean estimates $\hat{\mu}_1, \hat{\mu}_2, \ldots, \hat{\mu}_k$.
$p \leftarrow \frac{1}{2^{d/2}k \cdot \Gamma(d/2+1)} \cdot \max_{0 \leq r \leq \epsilon/2} r^d e^{-r^2/2}$;
$N \leftarrow (2\ln k)/p$; $C \leftarrow \emptyset$;
**for** $i = 1, 2, \ldots, N$ **do**
> Draw $X_i$ from $P$;
> Run Algorithm 2 with $\mu^* = X_i$ for $\Theta(\log N)$ times;
> **if** *Algorithm 2 accepts more than half of the times* **then**
> > $C \leftarrow C \cup \{X_i\}$;

**end**
Partition $C$ into $C_1, C_2, \ldots, C_k$ such that $x, y \in C$ are in the same cluster if $\|x - y\|_2 \leq 2\epsilon$;
Arbitrarily pick $\hat{\mu}_1 \in C_1, \hat{\mu}_2 \in C_2, \ldots, \hat{\mu}_k \in C_k$;
**return** $\hat{\mu}_1, \ldots, \hat{\mu}_k$;

---

**Algorithm 2:** Testing Gaussian Mixtures using Fourier Transform

---

**Input:** Sample access to mixture $P$. Parameters $\epsilon, \sigma, M > 0$ and candidate mean $\mu^* \in \mathbb{R}^d$.
**Output:** "Accept" or "Reject".
$\gamma \leftarrow (\sigma^2/2 + 1)\epsilon^2/64$;
$N \leftarrow \Theta((k/\gamma)^2) \cdot e^{\Theta(d+M^2)}$;
$\text{Avg} \leftarrow 0$;
**for** $i = 1, 2, \ldots, N$ **do**
> Draw $X \sim P$ and $\xi \sim \mathcal{N}(0, \sigma^2 I_d)$;
> **if** $\|\xi\|_2 \leq M$ **then**
> > $\text{Avg} \leftarrow \text{Avg} + \frac{2^{d/2}k}{N} \cdot e^{\|\xi\|_2^2/4} \cdot e^{-\|X-\mu^*\|_2^2/2} \cdot e^{i\xi^\top(X-\mu^*)}$;

**end**
$\theta \leftarrow \frac{1}{2}\left[e^{-(\sigma^2/2+1)\epsilon^2/16} + e^{-(\sigma^2/2+1)\epsilon^2/4}\right]$;
**return** "Accept" if $\text{Re Avg} \geq \theta$ and "Reject" otherwise;

---

# B   Auxiliary Lemmas

**Lemma B.1.** *For integers $n, k \geq 0$, $k! \geq e^{-k}k^k$ and $\binom{n}{k} \leq \left(\frac{en}{k}\right)^k$.*

*Proof.* By the Taylor expansion of $e^x$, $e^k = \sum_{j=0}^{+\infty} \frac{k^j}{j!} \geq \frac{k^j}{j!}\big|_{j=k} = \frac{k^k}{k!}$. Rearranging gives the first inequality. The second inequality then follows from $\binom{n}{k} = \frac{n(n-1)\cdots(n-k+1)}{k!} \leq \frac{n^k}{e^{-k}k^k} = \left(\frac{en}{k}\right)^k$. $\square$

We will need the following standard facts about the volume of an $\ell_2$-ball in high dimensions.

**Lemma B.2.** *The volume of a $d$-ball of radius $r$ is*

$$\text{Vol}_d(r) = \frac{\pi^{d/2}}{\Gamma(d/2+1)} \cdot r^d \leq (2r)^d.$$

*Furthermore, $\Gamma(d/2+1) = d^{O(d)}$.*

*Proof.* We refer the reader to Smith and Vamanamurthy [1989] for a proof of the identity. The inequality $\text{Vol}_d(r) \leq (2r)^d$ holds since the $\ell_2$ ball is contained in the $\ell_\infty$ ball of radius $r$, which trivially

has volume $(2r)^d$. Finally, the bound $\Gamma(d/2 + 1) = d^{O(d)}$ follows from Stirling's approximation $\Gamma(z) = (1 + O(1/z))\sqrt{2\pi/z}(z/e)^z$. $\qquad\square$

We need the following tail bound of $\chi^2$-distributions proved by Laurent and Massart [2000, Equation (4.3)], to control the probability that a Gaussian random variable has a large norm.

**Lemma B.3.** *Let random variable $X$ be sampled from the $\chi^2$-distribution with $d$ degrees of freedom. Then, for any $t > 0$,*

$$\Pr_X\left[X \geq d + 2\sqrt{dt} + 2t\right] \leq e^{-t}.$$

*Furthermore, for any $t \geq 5d$,*

$$\Pr_X[X \geq t] \leq e^{-t/5}.$$

The following classic theorem in topology is used in our lower bound proof. See, e.g., Matoušek et al. [2003] for a proof of the Borsuk–Ulam theorem. We remark that Borsuk–Ulam has been applied in a similar fashion in the literature of mixture learning [Hardt and Price, 2015, Chen et al., 2020].

**Theorem B.4** (The Borsuk–Ulam Theorem). *Suppose that $n \geq m$ and $f : \mathbb{S}^n \to \mathbb{R}^m$ is continuous. Then, there exists $x \in \mathbb{S}^n$ such that $f(x) = f(-x)$.*

## C Deferred Proofs from Section 2

*Proof of Theorem 2.1.* The proof proceeds by first generating many candidates for the means $\mu_1, \ldots, \mu_k$, and then verifying them using the tester from Theorem 2.3.

**Finding candidates.** We first show that if we draw sufficiently many samples from $P$, for every mean vector $\mu_j$ there is a sample that is $O(\epsilon)$-close to it. Indeed, for any $j \in [k]$, the probability that a sample from $P$ is $(\epsilon/2)$-close to $\mu_j$ is at least

$$
\begin{aligned}
\frac{1}{k} \cdot \Pr_{X \sim \mathcal{N}(\mu_j, I_d)}[\|X - \mu_j\|_2 \leq \epsilon/2] &\geq \frac{1}{k} \cdot \max_{0 \leq r \leq \epsilon/2}\left[\mathrm{Vol}_d(r) \cdot \frac{1}{(2\pi)^{d/2}}e^{-r^2/2}\right] \\
&= \frac{1}{k} \cdot \max_{0 \leq r \leq \epsilon/2}\left[\frac{\pi^{d/2}r^d}{\Gamma(d/2 + 1)} \cdot \frac{e^{-r^2/2}}{(2\pi)^{d/2}}\right] \quad \text{(Lemma B.2)} \\
&= \frac{1}{2^{d/2}k \cdot \Gamma(d/2 + 1)} \cdot \max_{0 \leq r \leq \epsilon/2} r^d e^{-r^2/2} =: p.
\end{aligned}
$$

Our algorithm first draws $N := (2\ln k)/p$ points $X_1, X_2, \ldots, X_N$ from $P$. The probability that none of them is $(\epsilon/2)$-close to $\mu_j$ is at most $(1 - p)^N \leq 1/k^2$. By a union bound, w.h.p. it holds for every $j \in [k]$ that some $X_i$ is $(\epsilon/2)$-close to $\mu_j$.

**Running the tester.** Then, we run the testing algorithm from Theorem 2.3 with $\mu^*$ set to each of the $N$ candidates $X_1, X_2, \ldots, X_N$. For each point $X_i$, we repeat the tester $\Theta(\log N)$ times (with new samples each time) and take the majority vote of the decisions. By a Chernoff bound, w.h.p. the decisions are simultaneously correct for all the $N$ points, i.e., for each $X_i$ the majority vote "accepts" if $\min_{j \in [k]} \|\mu_j - X_i\|_2 \leq \epsilon/2$ and "rejects" if this minimum distance is at least $\epsilon$.

**Clustering.** We focus on $C := \{i \in [N] : \text{majority vote accepts } X_i\}$ and define $C_j := \{i \in C : \|X_i - \mu_j\|_2 \leq \epsilon\}$ for $j \in [k]$. The correctness of the tester implies that for every $i \in C$, $X_i$ must be $\epsilon$-close to some mean vector $\mu_j$, so $\bigcup_{j \in [k]} C_j = C$. Furthermore, we claim that $\{C_j\}_{j \in [k]}$ are pairwise disjoint and thus constitute a partition of $C$. Suppose towards a contradiction that some $i \in C_j \cap C_{j'}$. Then, since $\epsilon < \Delta/100$, we get $\|\mu_j - \mu_{j'}\|_2 \leq \|\mu_j - X_i\|_2 + \|\mu_{j'} - X_i\|_2 \leq 2\epsilon < \Delta$, a contradiction. We also claim that every $C_j$ is non-empty. This is because we proved earlier that at least one of the $X_i$'s is $\epsilon/2$-close to $\mu_j$, and the majority vote must accept such an $X_i$.

It remains to show that we can easily identify such a partition $\{C_j\}$ without knowing $\mu_1, \ldots, \mu_k$. We note that for any $i \in C_j$ and $i' \in C_{j'}$. If $j = j'$, we have

$$\|X_i - X_{i'}\|_2 \leq \|X_i - \mu_j\|_2 + \|X_{i'} - \mu_j\|_2 \leq 2\epsilon.$$

If $j \neq j'$, the condition that $\epsilon < \Delta/100$ gives

$$\|X_i - X_{i'}\|_2 \geq \|\mu_j - \mu_{j'}\| - \|X_i - \mu_j\|_2 - \|X_{i'} - \mu_{j'}\|_2 \geq \Delta - 2\epsilon > 2\epsilon.$$

Therefore, if we cluster $C$ such that $i, i' \in C$ belong to the same cluster if and only if $\|X_i - X_{i'}\|_2 \leq 2\epsilon$, we obtain the exact partition $\{C_j\}$. We can then recover $\mu_1, \ldots, \mu_k$ up to an error of $\epsilon$ by outputting an arbitrary element in each of the $k$ clusters.

**Runtime.** To upper bound the runtime, we note that

$$N = (2 \ln k)/p = O(k \log k) \cdot 2^{d/2} \Gamma(d/2 + 1) \cdot \min_{0 \leq r \leq \epsilon/2} e^{r^2/2} r^{-d}$$

$$= O(k \log k) \cdot d^{O(d)} \cdot \begin{cases} (2/\epsilon)^d e^{\epsilon^2/8}, & \epsilon \leq 2\sqrt{d}, \\ (e/d)^{d/2}, & \epsilon > 2\sqrt{d} \end{cases} \quad \text{(Lemma B.2)}$$

$$= O(k \log k) \cdot \max\{(d/\epsilon)^{O(d)}, d^{O(d)}\}.$$

Therefore, the runtime of the algorithm is upper bounded by

$$O(N \log N) = O(k \log^2 k) \cdot \max\left\{(d/\epsilon)^{O(d)}, d^{O(d)}\right\}$$

times that of the tester from Theorem 2.3. This finishes the proof. $\square$

## D  Deferred Proofs from Section 3

We start with the proof of Lemma 3.1, which we restate below.

**Lemma 3.1** *For $P = \frac{1}{k} \sum_{j=1}^{k} \mathcal{N}(\mu_j, I_d)$ and any $\xi \in \mathbb{R}^d$,*

$$\mathop{\mathbb{E}}_{X \sim P}\left[e^{-\|X\|_2^2/2} \cdot e^{i(\xi^\top X)}\right] = \frac{e^{-\|\xi\|_2^2/4}}{2^{d/2}k} \sum_{j=1}^{k} e^{-\|\mu_j\|_2^2/4} \cdot e^{i(\mu_j^\top \xi/2)} = \frac{e^{-\|\xi\|_2^2/4}}{2^{d/2}k} \cdot A_\mu(\xi),$$

*where we define $A_\mu(\xi) := \sum_{j=1}^{k} e^{-\|\mu_j\|_2^2/4} \cdot e^{i(\mu_j^\top \xi/2)}$.*

*Proof of Lemma 3.1.* For any $v, \xi \in \mathbb{R}^d$, we have

$$\mathop{\mathbb{E}}_{X \sim \mathcal{N}(v, I_d)}\left[e^{-\|X\|_2^2/2} \cdot e^{i(\xi^\top X)}\right] = \mathop{\mathbb{E}}_{X \sim \mathcal{N}(v, I_d)}\left[\prod_{j=1}^{d} e^{-X_j^2/2} \cdot e^{i(\xi_j X_j)}\right]$$

$$= \prod_{j=1}^{d} \mathop{\mathbb{E}}_{X_j \sim \mathcal{N}(v_j, 1)}\left[e^{-X_j^2/2} \cdot e^{i(\xi_j X_j)}\right].$$

$$(X_1, \ldots, X_d \text{ are independent})$$

The $j$-th term in the product above is given by

$$\frac{1}{\sqrt{2\pi}} \int_{-\infty}^{+\infty} e^{-(x-v_j)^2/2 - x^2/2} \cdot e^{i\xi_j x} \, \mathrm{d}x = \frac{1}{\sqrt{2}} \cdot e^{-\xi_j^2/4} \cdot e^{-v_j^2/4} \cdot e^{iv_j\xi_j/2},$$

so we have

$$\mathop{\mathbb{E}}_{X \sim \mathcal{N}(v, I_d)}\left[e^{-\|X\|_2^2/2} \cdot e^{i(\xi^\top X)}\right] = \prod_{j=1}^{d}\left(\frac{1}{\sqrt{2}} \cdot e^{-\xi_j^2/4} \cdot e^{-v_j^2/4} \cdot e^{iv_j\xi_j/2}\right)$$

$$= \frac{e^{-\|\xi\|_2^2/4}}{2^{d/2}} \cdot e^{-\|v\|_2^2/4} \cdot e^{i(v^\top \xi/2)}.$$

Finally, the lemma follows from the above identity and averaging over $v \in \{\mu_1, \ldots, \mu_k\}$. $\square$

**Lemma 3.2** *For any $M, \sigma > 0$ that satisfy $M^2/\sigma^2 \geq 5d$,*

$$T_\mu = \sum_{j=1}^{k} e^{-(\sigma^2/2+1)\|\mu_j\|_2^2/4} + O\left(e^{-M^2/(5\sigma^2)}\right) \cdot \sum_{j=1}^{k} e^{-\|\mu_j\|_2^2/4},$$

*where the $O(x)$ notation hides a complex number with modulus $\leq x$.*

*Proof of Lemma 3.2.* Recall that $A_\mu(\xi) = \sum_{j=1}^{k} e^{-\|\mu_j\|_2^2/4} \cdot e^{i(\mu_j^\top \xi/2)}$. The contribution of the $j$-th term of $A_\mu(\xi)$ to $T_\mu$ is

$$\mathop{\mathbb{E}}_{\xi \sim \mathcal{N}(0,\sigma^2 I_d)} \left[ e^{-\|\mu_j\|_2^2/4} \cdot e^{i(\mu_j^\top \xi/2)} \cdot \mathbb{1}\left[\|\xi\|_2 \leq M\right] \right]$$

$$= e^{-\|\mu_j\|_2^2/4} \cdot \left[ \mathop{\mathbb{E}}_{\xi \sim \mathcal{N}(0,\sigma^2 I_d)} \left[ e^{i(\mu_j^\top \xi/2)} \right] + O\left( \mathop{\Pr}_{\xi \sim \mathcal{N}(0,\sigma^2 I_d)} \left[\|\xi\|_2 > M\right] \right) \right]$$

$$= e^{-\|\mu_j\|_2^2/4} \cdot \left[ e^{-\sigma^2\|\mu_j\|_2^2/8} + O\left(e^{-M^2/(5\sigma^2)}\right) \right] \qquad (M^2/\sigma^2 \geq 5d \text{ and Lemma B.3})$$

$$= e^{-(\sigma^2/2+1)\|\mu_j\|_2^2/4} + e^{-\|\mu_j\|_2^2/4} \cdot O\left(e^{-M^2/(5\sigma^2)}\right).$$

The lemma then follows from a summation over $j \in [k]$. $\qquad \square$

To prove Claims 3.3 and 3.4, we need the following lower bound on the norms of the mean vectors.

**Lemma D.1.** *Suppose that $\mu_1, \mu_2, \ldots, \mu_k \in \mathbb{R}^d$ are $\Delta$-separated, and $\|\mu_1\|_2 \leq \|\mu_2\|_2 \leq \cdots \leq \|\mu_k\|_2$. For any $j \geq 2$,*

$$\|\mu_j\|_2 \geq \max\left\{ \frac{\Delta}{2}, \frac{\Delta j^{1/d}}{4} \right\}.$$

*Proof of Lemma D.1.* Fix $j \geq 2$. The first lower bound follows from

$$\Delta \leq \|\mu_1 - \mu_j\|_2 \leq \|\mu_1\|_2 + \|\mu_j\|_2 \leq 2\|\mu_j\|_2.$$

Now we turn to the second bound. Since $\mu_1, \ldots, \mu_j$ are $\Delta$-separated, the $j$ balls $B(\mu_1, \Delta/2)$, $B(\mu_2, \Delta/2)$, ..., $B(\mu_j, \Delta/2)$ are disjoint and all contained in $B(0, \|\mu_j\|_2 + \Delta/2)$. Thus, we have $j \cdot (\Delta/2)^d \leq (\|\mu_j\|_2 + \Delta/2)^d$, which implies $\|\mu_j\|_2 \geq \frac{\Delta}{2} \cdot (j^{1/d} - 1)$.

For $j \geq 2^d$, we have $j^{1/d}/2 \geq 1$ and the lower bound can be relaxed to

$$\|\mu_j\|_2 \geq \frac{\Delta}{2} \cdot \left( j^{1/d} - \frac{j^{1/d}}{2} \right) = \frac{\Delta j^{1/d}}{4}.$$

For $2 \leq j < 2^d$, we have $j^{1/d} < 2$, so the first lower bound implies $\|\mu_j\|_2 \geq \frac{\Delta}{2} \geq \frac{\Delta j^{1/d}}{4}$. $\qquad \square$

With the lower bounds on $\|\mu_j\|_2$, we are ready to bound $S_1 = \sum_{j=2}^{k} e^{-(\sigma^2/2+1)\|\mu_j\|_2^2/4}$ and $S_2 = \sum_{j=1}^{k} e^{-\|\mu_j\|_2^2/4}$. We first restate and prove Claim 3.3:

**Claim 3.3** *Assuming $(\sigma^2/2 + 1)\Delta^2 \geq 100 \min\{\ln k, d\}$,*

$$S_1 \leq 2e^{-(\sigma^2/2+1)\Delta^2/64} \cdot \min\{k, 2^d\}.$$

*Proof of Claim 3.3.* If $k \leq 2^d$, we apply the bound $\|\mu_j\|_2 \geq \Delta/2$ to all $j \geq 2$ and get $S_1 \leq (k-1) \cdot e^{-(\sigma^2/2+1)\Delta^2/16}$, which is stronger than what we need. Otherwise, we apply $\|\mu_j\|_2 \geq \Delta j^{1/d}/4$

and get

$$S_1 \leq \sum_{j=2}^{k} \exp\left(-\frac{(\sigma^2/2+1)\Delta^2 j^{2/d}}{64}\right)$$

$$\leq \sum_{j=2}^{k} \exp\left(-\frac{(\sigma^2/2+1)\Delta^2 \lfloor j^{1/d}\rfloor^2}{64}\right) \qquad\qquad (j^{1/d} \geq \lfloor j^{1/d}\rfloor)$$

$$= \sum_{t=1}^{+\infty} \exp\left(-\frac{(\sigma^2/2+1)\Delta^2 t^2}{64}\right) \cdot \sum_{j=2}^{k} \mathbb{1}\left[\lfloor j^{1/d}\rfloor = t\right]$$

$$\leq \sum_{t=1}^{+\infty} \exp\left(-\frac{(\sigma^2/2+1)\Delta^2 t^2}{64}\right) \cdot (t+1)^d. \qquad (\lfloor j^{1/d}\rfloor = t \implies j < (t+1)^d)$$

In the last summation, the ratio between the $(t+1)$-th term and the $t$-th term is given by

$$\exp\left(-\frac{(\sigma^2/2+1)\Delta^2(2t+1)}{64}\right) \cdot \left(1+\frac{1}{t+1}\right)^d \leq \exp\left(-\frac{3(\sigma^2/2+1)\Delta^2}{64}\right) \cdot \left(\frac{3}{2}\right)^d,$$

which is smaller than $1/2$ under the assumption that $(\sigma^2/2+1)\Delta^2 \geq 100\min\{\ln k, d\}$. Thus, the summation is at most twice the first term, i.e., $S_1 \leq 2 \cdot e^{-(\sigma^2/2+1)\Delta^2/64} \cdot 2^d$. $\qquad\square$

Now we restate and prove Claim 3.4:

**Claim 3.4** *We have*

$$S_2 \leq \begin{cases} 2, & \Delta^2 \geq 100d, \\ 1 + \frac{267d}{\Delta^2} \cdot \max\left\{\left(\frac{32d}{\Delta^2}\right)^{d/2}, 1\right\}, & \Delta^2 < 100d. \end{cases}$$

*Furthermore, $S_2 \leq 10 \cdot \min\left\{k, 1 + \left(\frac{32d}{\Delta^2}\right)^{d/2+1}\right\}$.*

*Proof of Claim 3.4.* Using $\|\mu_j\|_2 \geq \Delta j^{1/d}/4$ for $j \geq 2$, we can upper bound $S_2$ as follows:

$$S_2 = \sum_{j=1}^{k} e^{-\|\mu_j\|_2^2/4} \leq 1 + \sum_{j=2}^{k} \exp\left(-\frac{\Delta^2 j^{2/d}}{64}\right)$$

$$\leq 1 + \sum_{j=2}^{k} \exp\left(-\frac{\Delta^2 \lfloor j^{2/d}\rfloor}{64}\right) \qquad\qquad (j^{2/d} \geq \lfloor j^{2/d}\rfloor)$$

$$= 1 + \sum_{t=1}^{\lfloor k^{2/d}\rfloor} \exp(-\Delta^2 t/64) \cdot \sum_{j=2}^{k} \mathbb{1}\left[\lfloor j^{2/d}\rfloor = t\right]$$

$$\leq 1 + \sum_{t=1}^{\lfloor k^{2/d}\rfloor} \exp(-\Delta^2 t/64) \cdot (t+1)^{d/2}. \quad (\lfloor j^{2/d}\rfloor = t \implies j < (t+1)^{d/2})$$

In the last summation, the ratio between the $(t+1)$-th term and the $t$-th term is given by

$$R_t := \frac{\exp(-\Delta^2(t+1)/64) \cdot (t+2)^{d/2}}{\exp(-\Delta^2 t/64) \cdot (t+1)^{d/2}} = e^{-\Delta^2/64} \cdot \left(1+\frac{1}{t+1}\right)^{d/2}.$$

**The first case.** When $\Delta^2 \geq 100d$, for every $t \geq 1$ we have

$$R_t \leq \exp\left(-\frac{100d}{64}\right) \cdot \left(\frac{3}{2}\right)^{d/2} \leq \left(e^{-25/16} \cdot \sqrt{\frac{3}{2}}\right)^d < 1/2.$$

This implies that the summation $\sum_{t=1}^{\lfloor k^{2/d}\rfloor} \exp(-\Delta^2 t/64) \cdot (t+1)^{d/2}$ is dominated by twice its first term, i.e.,

$$S_2 \leq 1 + 2 \cdot e^{-\Delta^2/64} \cdot 2^{d/2} \leq 1 + 2 \cdot \left(e^{-25/16} \cdot \sqrt{2}\right)^d < 2.$$

**The second case.** By a straightforward calculation,

$$R_t \le e^{-\Delta^2/128} \iff 1 + \frac{1}{t+1} \le \exp\left(\frac{\Delta^2}{64d}\right) \impliedby e^{\frac{1}{t+1}} \le \exp\left(\frac{\Delta^2}{64d}\right) \iff t+1 \ge \frac{64d}{\Delta^2}.$$

In other words, the terms in $\sum_{t=1}^{\lfloor k^{2/d} \rfloor} \exp(-\Delta^2 t/64) \cdot (t+1)^{d/2}$ start to decay at a rate of at least $e^{\Delta^2/128}$ after the first $O(d/\Delta^2)$ terms. Therefore, we have

$$S_2 \le 1 + \left(\frac{64d}{\Delta^2} + \frac{1}{1 - e^{-\Delta^2/128}}\right) \cdot \max_{t \ge 1}\left[\exp(-\Delta^2 t/64) \cdot (t+1)^{d/2}\right].$$

Using the inequality $1 - e^{-x} \ge \frac{e-1}{e} \cdot \min\{x, 1\}$ for $x \ge 0$, we have

$$\frac{1}{1 - e^{-\Delta^2/128}} \le \frac{e}{e-1} \cdot \max\left\{\frac{128}{\Delta^2}, 1\right\}$$

$$\le \frac{e}{e-1} \cdot \max\left\{\frac{128d}{\Delta^2}, \frac{100d}{\Delta^2}\right\} < \frac{203d}{\Delta^2}. \qquad (1 \le d \text{ and } \Delta^2 < 100d)$$

On the other hand, elementary calculus shows that the function $t \mapsto e^{-\Delta^2 t/64} \cdot (t+1)^{d/2}$ defined over $[0, +\infty)$ is maximized at $t^* = \frac{32d}{\Delta^2} - 1$ if $\frac{32d}{\Delta^2} \ge 1$, and at $t^* = 0$ otherwise. In either case, the maximum value is upper bounded by $\max\{\left(\frac{32d}{\Delta^2}\right)^{d/2}, 1\}$. Therefore, we conclude that in the second case,

$$S_2 \le 1 + \frac{267d}{\Delta^2} \cdot \max\left\{\left(\frac{32d}{\Delta^2}\right)^{d/2}, 1\right\}.$$

**The "furthermore" part.** If $\frac{d}{\Delta^2} \le \frac{1}{100}$, the first case implies that

$$S_2 \le 2 \le 10 \cdot \left[1 + \left(\frac{32d}{\Delta^2}\right)^{d/2+1}\right].$$

If $\frac{d}{\Delta^2} \in (\frac{1}{100}, \frac{1}{32}]$, the bound for the second case reduces to

$$S_2 \le 1 + \frac{267d}{\Delta^2} \le 1 + \frac{267}{32} < 10 \cdot \left[1 + \left(\frac{32d}{\Delta^2}\right)^{d/2+1}\right].$$

If $\frac{d}{\Delta^2} > \frac{1}{32}$, the bound for the second case implies

$$S_2 \le 1 + \frac{267}{32} \cdot \left(\frac{32d}{\Delta^2}\right)^{d/2+1} < 10 \cdot \left[1 + \left(\frac{32d}{\Delta^2}\right)^{d/2+1}\right].$$

Finally, the "furthermore" part follows from the above and the observation that $k$ is a trivial upper bound on $S_2$. $\qquad\square$

## E  Proof of Lower Bound

We first state the formal version of Theorem 1.2:

**Theorem E.1.** *Suppose that $k \ge 3$, $d \le \frac{\ln k}{\ln \ln k}$, $C \ge 100$, and $\ln(8eC) \le (1 - 1/e)\frac{\ln k}{d}$. Then, there are two mixtures $\tilde{P}$ and $\tilde{Q}$ of $k$ spherical Gaussians in $\mathbb{R}^d$, such that for some $\Delta = \Theta\left(\frac{d}{\sqrt{C \log k}}\right)$:*

- *Both $\tilde{P}$ and $\tilde{Q}$ are $\Delta$-separated.*

- *The means of $\tilde{P}$ and $\tilde{Q}$ are not $(\Delta/2)$-close.*

- *The total variation distance between $\tilde{P}$ and $\tilde{Q}$ satisfies $d_{TV}(\tilde{P}, \tilde{Q}) \le 2k^{-C}$.*

Setting $\omega(1) = C = k^{o(1/d)}$ in Theorem E.1 shows that with a separation of $\Delta = o(d/\sqrt{\log k})$, $k^{\omega(1)}$ samples are needed to recover the means up to an $O(\Delta)$ error.

The proof of Theorem E.1 applies standard moment matching techniques in the literature [Hardt and Price, 2015, Regev and Vijayaraghavan, 2017]. The proof proceeds by first constructing two sets of $\le k$ points in $\mathbb{R}^d$ that have the same lower order moments, and then showing that these matching moments imply that their convolutions with a standard Gaussian are close in TV-distance.

The following lemma states that we can choose the centers of two mixtures $\tilde{P}$ and $\tilde{Q}$ such that their low-degree mean moments are identical, whereas their parameters are $\Delta$-apart from each other.

**Lemma E.2.** *Suppose that $R > \Delta > 0$ and $\left[\frac{e(t+d)}{d}\right]^d \le N \le \left(\frac{R}{3\Delta}\right)^d$. There exist $2N$ points $\mu_1^{(P)}, \ldots, \mu_N^{(P)}, \mu_1^{(Q)}, \ldots, \mu_N^{(Q)}$ in $B(0, 2R)$ such that:*

- $\left\|\mu_i^{(P)} - \mu_j^{(P)}\right\|_2 \ge \Delta$ *and* $\left\|\mu_i^{(Q)} - \mu_j^{(Q)}\right\|_2 \ge \Delta$ *for $i \ne j$.*

- *For any permutation $\pi$ over $[N]$,* $\max_{i \in [N]} \left\|\mu_i^{(P)} - \mu_{\pi(i)}^{(Q)}\right\|_2 \ge \Delta.$

- *For any $t' \in [t]$,* $\frac{1}{N} \sum_{i=1}^{N} \left[\mu_i^{(P)}\right]^{\otimes t'} = \frac{1}{N} \sum_{i=1}^{N} \left[\mu_i^{(Q)}\right]^{\otimes t'}.$

*Proof of Lemma E.2.* Let $S$ be an arbitrary maximal $(3\Delta)$-separated subset of $B(0, R)$. Then, the collection $\{B(\mu, 3\Delta) : \mu \in S\}$ must cover the ball $B(0, R)$; otherwise, we could add to $S$ the point that is not covered. This implies $|S| \ge \frac{\text{Vol}_d(R)}{\text{Vol}_d(3\Delta)} = (\frac{R}{3\Delta})^d \ge N$, so we can choose $N$ arbitrary points $\mu_1, \mu_2, \ldots, \mu_N$ from $S$.

In the following, we construct the point sets $\{\mu_i^{(P)}\}$ and $\{\mu_i^{(Q)}\}$ by slightly perturbing $\{\mu_1, \ldots, \mu_N\}$. Let $M = \binom{t+d}{d}$. For any $t' \ge 1$, the degree-$t'$ moment tensor in $d$ dimensions has exactly $\binom{d+t'-1}{d-1}$ distinct entries. So, among the first $t$ moment tensors, the total number of distinct entries is

$$\binom{d}{d-1} + \binom{d+1}{d-1} + \cdots + \binom{d+t-1}{d-1} = \binom{d+t}{d} - 1 = M - 1.$$

We define a function $f : \mathbb{S}^{Nd-1} \to \mathbb{R}^{M-1}$ as follows. Given $x \in \mathbb{S}^{Nd-1}$, we group the $Nd$ coordinates of $x$ into $N$ groups, and view them as $N$ points $x_1, x_2, \ldots, x_N \in \mathbb{R}^d$. Let $c(x) := \frac{\Delta}{\max_{i \in [N]} \|x_i\|_2}$ and $\mu_i^{(x)} := \mu_i + c(x)x_i$. ($c(x)$ is well-defined, since $x \in \mathbb{S}^{Nd-1}$ guarantees that some $x_i$ is non-zero.) Finally, $f(x) \in \mathbb{R}^{M-1}$ is defined as the concatenation of the $M - 1$ entries in the first $t$ moment tensors of the uniform distribution over $\{\mu_i^{(x)}\}_{i \in [N]}$.

It can be easily verified that $f$ is continuous. Furthermore, our assumption that $\left[\frac{e(t+d)}{d}\right]^d \le N$ together with Lemma B.1 implies

$$Nd - 1 \ge N - 1 \ge \left[\frac{e(t+d)}{d}\right]^d - 1 \ge \binom{t+d}{d} - 1 = M - 1.$$

Thus, by the Borsuk–Ulam theorem (Theorem B.4), there exists $x \in \mathbb{S}^{Nd-1}$ such that $f(x) = f(-x)$.

We prove the lemma by setting $\mu_i^{(P)} = \mu_i^{(x)}$ and $\mu_i^{(Q)} = \mu_i^{(-x)}$. By definition of $c(x)$, we have $\|\mu_i^{(P)} - \mu_i\|_2 = \|\mu_i^{(Q)} - \mu_i\|_2 = c(x)\|x_i\|_2 \le \Delta$ for every $i \in [N]$. Thus, $\|\mu_i^{(P)}\|_2 \le \|\mu_i\|_2 + \|\mu_i^{(P)} - \mu_i\|_2 \le R + \Delta \le 2R$ and similarly $\mu_i^{(Q)} \in B(0, 2R)$ for every $i \in [N]$. Furthermore, for any $i \ne j$,

$$\|\mu_i^{(P)} - \mu_j^{(P)}\|_2 \ge \|\mu_i - \mu_j\|_2 - \|\mu_i^{(P)} - \mu_i\|_2 - \|\mu_j^{(P)} - \mu_j\|_2 \ge 3\Delta - \Delta - \Delta = \Delta,$$

and similarly $\|\mu_i^{(Q)} - \mu_j^{(Q)}\|_2 \ge \Delta$. This proves the first condition.

Moreover, we note that for $i^* \in \arg\max_{i \in [N]} \|x_i\|_2$, it holds that $c(x)\|x_{i^*}\|_2 = \Delta$. Thus, $\|\mu_{i^*}^{(P)} - \mu_{i^*}^{(Q)}\|_2 = 2c(x)\|x_{i^*}\|_2 = 2\Delta$. On the other hand, for any $j \neq i^*$,

$$\|\mu_{i^*}^{(P)} - \mu_j^{(Q)}\|_2 \geq \|\mu_{i^*} - \mu_j\|_2 - \|\mu_{i^*}^{(P)} - \mu_{i^*}\|_2 - \|\mu_j^{(Q)} - \mu_j\|_2 \geq 3\Delta - \Delta - \Delta = \Delta,$$

so $\mu_{i^*}^{(P)}$ cannot be matched to any $\mu_j^{(Q)}$ without incurring an error of $\Delta$. This proves the second condition.

Finally, the third condition follows from $f(x) = f(-x)$ and our definition of $f$. □

The following lemma allows us to relate the total variation distance between $\tilde{P}$ and $\tilde{Q}$ to their $\ell_2$ distance, which is more Fourier-friendly.

**Lemma E.3.** *Let $\tilde{P}$ and $\tilde{Q}$ be two mixtures of spherical Gaussians with all means contained in $B(0, 2R)$. Then, for any $\epsilon \in (0,1)$ and $R' = 2R + \sqrt{d} + \sqrt{2\ln(1/\epsilon)}$,*

$$d_{TV}(\tilde{P}, \tilde{Q}) \leq \epsilon + \frac{\sqrt{\mathrm{Vol}_d(R')}}{2} \|\tilde{P} - \tilde{Q}\|_2,$$

*where $\mathrm{Vol}_d(r)$ denotes the volume of a $d$-ball with radius $r$.*

*Proof of Lemma E.3.* By definition of the TV-distance,

$$
\begin{aligned}
d_{\mathrm{TV}}(\tilde{P}, \tilde{Q}) &= \frac{1}{2} \int \left| \tilde{P}(x) - \tilde{Q}(x) \right| \, dx \\
&\leq \frac{1}{2} \int_{\|x\|_2 \geq R'} \left[ \tilde{P}(x) + \tilde{Q}(x) \right] \, dx + \frac{1}{2} \int_{\|x\|_2 \leq R'} \left| \tilde{P}(x) - \tilde{Q}(x) \right| \, dx.
\end{aligned}
\tag{2}
$$

To bound the first term above, we note that $\int_{\|x\|_2 \geq R'} \tilde{P}(x) \, dx$ is exactly $\tilde{P}(\mathbb{R}^d \setminus B(0, R'))$, the probability that a sample from $\tilde{P}$ has norm greater than $R'$. Since the mean of every cluster of $\tilde{P}$ is contained in $B(0, 2R)$, this probability is upper bounded by the probability that a standard Gaussian random variable has norm $\geq R' - 2R$:

$$
\begin{aligned}
\tilde{P}\left( \mathbb{R}^d \setminus B(0, R') \right) &\leq \Pr_{X \sim N(0, I_d)} \left[ \|X\|_2 \geq R' - 2R \right] \\
&= \Pr_{X \sim \chi^2(d)} \left[ X \geq (\sqrt{d} + \sqrt{2\ln(1/\epsilon)})^2 \right] \\
&\leq \Pr_{X \sim \chi^2(d)} \left[ X \geq d + 2\sqrt{d\ln(1/\epsilon)} + 2\ln(1/\epsilon) \right] \\
&\leq \epsilon. \qquad \text{(Lemma B.3)}
\end{aligned}
$$

Similarly, we have $\tilde{Q}\left( \mathbb{R}^d \setminus B(0, R') \right) \leq \epsilon$, so the first term of Equation (2) is upper bounded by $\epsilon$.

The second term of Equation (2) can be bounded in terms of the $\ell_2$ distance between $\tilde{P}$ and $\tilde{Q}$ using Cauchy-Schwarz:

$$
\begin{aligned}
\int_{\|x\|_2 \leq R'} \left| \tilde{P}(x) - \tilde{Q}(x) \right| \, dx &\leq \sqrt{\int_{\|x\|_2 \leq R'} \left[ \tilde{P}(x) - \tilde{Q}(x) \right]^2 \, dx} \cdot \sqrt{\int_{\|x\|_2 \leq R'} 1 \, dx} \\
&\leq \|\tilde{P} - \tilde{Q}\|_2 \cdot \sqrt{\mathrm{Vol}_d(R')}.
\end{aligned}
$$

Plugging the above into Equation (2) proves $d_{\mathrm{TV}}(\tilde{P}, \tilde{Q}) \leq \epsilon + \frac{\sqrt{\mathrm{Vol}_d(R')}}{2} \|\tilde{P} - \tilde{Q}\|_2$. □

The following lemma upper bounds the $\ell_2$-distance between $\tilde{P}$ and $\tilde{Q}$ under the assumption that their low-degree moments are equal.

**Lemma E.4.** *Suppose that $t/(4R) \geq \sqrt{5d}$, the supports of $P$ and $Q$ are contained in $B(0, 2R)$, and the first $t$ moment tensors of $P$ and $Q$ are equal. Let $\tilde{P} = P * \mathcal{N}(0, I_d)$ and $\tilde{Q} = Q * \mathcal{N}(0, I_d)$. We have*

$$\|\tilde{P} - \tilde{Q}\|_2^2 \leq 4\exp\left( -\frac{t^2}{80R^2} \right) + 2\left( \frac{t}{4R} \right)^d \cdot \frac{(2R)^{2t}}{t!}.$$

*Proof of Lemma E.4.* By the Plancherel theorem,

$$\|\tilde{P} - \tilde{Q}\|_2^2 = \frac{1}{(2\pi)^d} \int \left| (\mathcal{F}\tilde{P})(\xi) - (\mathcal{F}\tilde{Q})(\xi) \right|^2 \, \mathrm{d}\xi,$$

where $(\mathcal{F}\tilde{P})(\xi) := \int \tilde{P}(x) e^{i\xi^\top x} \, \mathrm{d}x$ and $(\mathcal{F}\tilde{Q})(\xi) := \int \tilde{Q}(x) e^{i\xi^\top x} \, \mathrm{d}x$.[5]

Since $\tilde{P}$ is the convolution of $P$ and $\mathcal{N}(0, I_d)$, we have

$$(\mathcal{F}\tilde{P})(\xi) = (\mathcal{F}\mathcal{N}(0, I_d))(\xi) \cdot (\mathcal{F}P)(\xi) = e^{-\|\xi\|_2^2/2} \mathop{\mathbb{E}}_{\mu \sim P} \left[ e^{i\xi^\top \mu} \right].$$

Similarly, $(\mathcal{F}\tilde{Q})(\xi) = e^{-\|\xi\|_2^2/2} \mathbb{E}_{\mu \sim Q} \left[ e^{i\xi^\top \mu} \right]$ and thus,

$$\|\tilde{P} - \tilde{Q}\|_2^2 = \frac{1}{(2\pi)^d} \int e^{-\|\xi\|_2^2} \left| \mathop{\mathbb{E}}_{\mu \sim P} \left[ e^{i\xi^\top \mu} \right] - \mathop{\mathbb{E}}_{\mu \sim Q} \left[ e^{i\xi^\top \mu} \right] \right|^2 \, \mathrm{d}\xi. \tag{3}$$

Since the first $t$ moments tensors of $P$ and $Q$ are equal, for any $\xi \in \mathbb{R}^d$,

$$\left| \mathop{\mathbb{E}}_{\mu \sim P} \left[ e^{i\xi^\top \mu} \right] - \mathop{\mathbb{E}}_{\mu \sim Q} \left[ e^{i\xi^\top \mu} \right] \right|$$

$$= \left| \sum_{j=0}^{+\infty} \frac{i^j}{j!} \left[ \mathop{\mathbb{E}}_{\mu \sim P} [(\xi^\top \mu)^j] - \mathop{\mathbb{E}}_{\mu \sim Q} [(\xi^\top \mu)^j] \right] \right| \qquad \text{(Taylor expansion)}$$

$$\le \sum_{j=0}^{+\infty} \frac{1}{j!} \left| \mathop{\mathbb{E}}_{\mu \sim P} [(\xi^\top \mu)^j] - \mathop{\mathbb{E}}_{\mu \sim Q} [(\xi^\top \mu)^j] \right| \qquad \text{(triangle inequality)}$$

$$\le \sum_{j=t+1}^{+\infty} \frac{1}{j!} \left| \mathop{\mathbb{E}}_{\mu \sim P} [(\xi^\top \mu)^j] - \mathop{\mathbb{E}}_{\mu \sim Q} [(\xi^\top \mu)^j] \right| \qquad \text{(identical first $t$ moments)}$$

$$\le \sum_{j=t+1}^{+\infty} \frac{2}{j!} (2R\|\xi\|_2)^j. \qquad \text{(Cauchy-Schwarz)}$$

If $\|\xi\|_2 \le t/(4R)$, the last summation above can be upper bounded by $2 \cdot \frac{(2R\|\xi\|_2)^t}{t!}$. Otherwise, we can use the trivial upper bound $\left| \mathbb{E}_{\mu \sim P} \left[ e^{i\xi^\top \mu} \right] - \mathbb{E}_{\mu \sim Q} \left[ e^{i\xi^\top \mu} \right] \right| \le 2$. Plugging these back to Equation (3) gives

$$\|\tilde{P} - \tilde{Q}\|_2^2 \le \frac{1}{(2\pi)^d} \int_{\|\xi\|_2 \ge t/(4R)} 4e^{-\|\xi\|_2^2} \, \mathrm{d}\xi + \frac{1}{(2\pi)^d} \int_{\|\xi\|_2 \le t/(4R)} 4e^{-\|\xi\|_2^2} \cdot \frac{(2R\|\xi\|_2)^{2t}}{(t!)^2} \, \mathrm{d}\xi.$$

Using Lemma B.3 and the assumption that $t/(4R) \ge \sqrt{5d}$, the first term above is upper bounded by

$$4 \cdot \frac{1}{(2\pi)^{d/2}} \int_{\|x\|_2 \ge t/(4R)} e^{-\|x\|_2^2/2} \, \mathrm{d}x = 4 \mathop{\Pr}_{X \sim \mathcal{N}(0, I_d)} [\|X\|_2 \ge t/(4R)] \le 4 \exp\left( -\frac{t^2}{80R^2} \right).$$

To bound the second term, we note that the function $x \mapsto e^{-x} x^t$ is maximized at $x = t$. Thus, the second term is upper bounded by

$$\frac{1}{(2\pi)^d} \cdot \mathrm{Vol}_d(t/(4R)) \cdot 4e^{-t} \frac{(2R)^{2t} t^t}{(t!)^2} \le 2 \left( \frac{t}{4R} \right)^d \cdot \frac{(2R)^{2t}}{t!}.$$

Here we use $2^{-d} \mathrm{Vol}_d(r) \le r^d$ (Lemma B.2), $4/\pi^d < 2$, and $e^{-t} t^t \le t!$ (Lemma B.1). $\qquad \square$

Now we are ready to prove Theorem E.1 by combining Lemmas E.2 through E.4 with carefully chosen parameters.

---

[5]Here the $\frac{1}{(2\pi)^d}$ factor comes from that our definition of the Fourier transform differs from the standard one by a factor of $2\pi$ on the exponent.

*Proof of Theorem E.1.* Let $R > \Delta > 0$ and integers $N, t$ be parameters to be chosen later. Assuming that $\left[\frac{e(t+d)}{d}\right]^d \le N \le \left(\frac{R}{3\Delta}\right)^d$, by Lemma E.2, there exist two distributions $P$ and $Q$ such that: (1) $P$ (resp. $Q$) is the uniform distribution over $N$ points $\mu_1^{(P)}, \dots, \mu_N^{(P)}$ (resp. $\mu_1^{(Q)}, \dots, \mu_N^{(Q)}$) that are $\Delta$-separated and contained in $B(0, 2R)$; (2) $P$ and $Q$ have the same first $t$ moments; (3) the two mixtures $\tilde{P} = P * \mathcal{N}(0, I_d)$ and $\tilde{Q} = Q * \mathcal{N}(0, I_d)$ are $\Delta$-far from each other in their parameters.

Then, applying Lemmas E.3 and E.4 to $\tilde{P}$ and $\tilde{Q}$ gives

$$d_{\mathrm{TV}}(\tilde{P}, \tilde{Q}) \le \epsilon + \frac{\sqrt{\mathrm{Vol}_d(R')}}{2} \cdot \sqrt{4\exp\left(-\frac{t^2}{80R^2}\right) + 2\left(\frac{t}{4R}\right)^d \cdot \frac{(2R)^{2t}}{t!}}.$$

We will set $\epsilon = k^{-C}$ and ensure the following:

$$\sqrt{\mathrm{Vol}_d(R')} \le 1/\epsilon, \quad \exp\left(-\frac{t^2}{80R^2}\right) \le \epsilon^4/2, \quad \left(\frac{t}{4R}\right)^d \cdot \frac{(2R)^{2t}}{t!} \le \epsilon^4.$$

These together imply $d_{\mathrm{TV}}(\tilde{P}, \tilde{Q}) \le 2\epsilon = 2k^{-C}$ as desired.

**Choice of parameters.** With some calculation, we can show that the above conditions can be satisfied by setting $t = 4C\ln k$ and $R = \frac{\sqrt{C\ln k}}{10}$. Now we set the separation $\Delta$ so that $\left[\frac{e(t+d)}{d}\right]^d \le N \le \left(\frac{R}{3\Delta}\right)^d$ could hold. Since $d \le \ln k \le t$, it is sufficient to guarantee $\frac{2et}{d} \le \frac{R}{3\Delta}$, which holds for $\Delta = \frac{Rd}{6et} = \frac{d}{240e\sqrt{C\ln k}} = \Theta\left(\frac{d}{\sqrt{C\log k}}\right)$.

It remains to verify a few additional assumptions that are required for applying the lemmas. First, our choice of $N$ must be at most $k$. This is equivalent to $k \ge \left(\frac{2et}{d}\right)^d = \left(\frac{8eC\ln k}{d}\right)^d$. Since for any $d > 0$,

$$\left(\frac{\ln k}{d}\right)^d = \exp(d\ln\ln k - d\ln d) \le \exp(d\ln\ln k - d\ln d)|_{d=e^{-1}\ln k} = k^{1/e},$$

it is then sufficient to have $(8eC)^d \le k^{1-1/e}$, but this is guaranteed by the assumption that $\ln(8eC) \le (1 - 1/e)\frac{\ln k}{d}$. Second, we need to verify that $R > \Delta$. This is equivalent to $24eC\ln k > d$, which clearly holds given $C \ge 100$ and $d \le \frac{\ln k}{\ln\ln k}$. Finally, we need $t/(4R) \ge \sqrt{5d}$ to apply Lemma E.4. This inequality is equivalent to $20C\ln k \ge d$, which also clearly holds.

**Detailed calculation.** We first show that $\sqrt{\mathrm{Vol}_d(R')} \le 1/\epsilon$. Recall that $R' = 2R + \sqrt{d} + \sqrt{2\ln(1/\epsilon)} = 2R + \sqrt{d} + \sqrt{2C\ln k}$. We have

$$\ln\sqrt{\mathrm{Vol}_d(R')} \le \frac{d}{2}\ln(4R + 2\sqrt{d} + 2\sqrt{2C\ln k}) \qquad \text{(Lemma B.2)}$$

$$\le \frac{d}{2}\ln(16R\sqrt{2dC\ln k}) \qquad (x, y, z \ge 2 \implies x + y + z \le xyz)$$

$$\le \frac{\ln k}{2\ln\ln k} \cdot \ln\left[\frac{16\sqrt{2}}{10}\sqrt{C\ln k} \cdot \sqrt{\frac{C\ln^2 k}{\ln\ln k}}\right] \qquad \text{(choice of } R \text{ and } d \le \frac{\ln k}{\ln\ln k})$$

$$\le \frac{\ln k}{2\ln\ln k} \cdot \ln(eC) + \frac{\ln k}{2\ln\ln k} \cdot \frac{3}{2}\ln\ln k \qquad (16\sqrt{2}/10 < e)$$

$$\le \frac{C\ln k}{2} + \frac{3C\ln k}{400} < C\ln k. \qquad (C \ge 100)$$

This proves $\sqrt{\mathrm{Vol}_d(R')} \le k^C = 1/\epsilon$.

Then, we ensure that $\exp\left(-\frac{t^2}{80R^2}\right) \le \epsilon^4/2 = k^{-4C}/2$. This is equivalent to $R \le \frac{t}{\sqrt{80\ln(2k^{4C})}}$, which holds given our choice of $R = \frac{t}{40\sqrt{C\ln k}}$.

Finally, we deal with the constraint that $\left(\frac{t}{4R}\right)^d \cdot \frac{(2R)^{2t}}{t!} \leq \epsilon^4 = k^{-4C}$. After taking a logarithm on both sides, the above inequality reduces to

$$d \ln t + 2t \ln(2R) + 4C \ln k \leq d \ln(4R) + \ln(t!).$$

Since $\ln(t!) \geq t \ln t - t$ (Lemma B.1) and $t = 4C \ln k$, it is sufficient to have

$$2t \ln(2R) + d \ln t + 2t \leq t \ln t.$$

We will show that in the left-hand side above, the first term is at most $t \ln t - 3t$, while the second terms is bounded by $t$. This would finish the proof. Indeed, we have

$$2t \ln(2R) = t \ln(2R)^2 = t \ln \frac{C \ln k}{25} \leq t \ln \frac{t}{e^3} = t \ln t - 3t.$$

For the second term, we have

$$d \ln t \leq \frac{\ln k}{\ln \ln k} \cdot \ln(4C \ln k) \leq \ln k \cdot \ln(4C) + \ln k < 4C \ln k = t,$$

so $d \ln t \leq t$ holds. This completes the proof of the theorem. $\qquad\square$

# F    Extension to Non-Gaussian Mixtures

Given a probability distribution $\mathcal{D}$ over $\mathbb{R}^d$, the location family defined by $\mathcal{D}$ is

$$\{\mathcal{D}_\mu : \mu \in \mathbb{R}^d\},$$

where each $\mathcal{D}_\mu$ is the distribution of $X + \mu$ when $X$ is drawn from $\mathcal{D}$. In other words, $\mathcal{D}_\mu$ is obtained by translating the distribution $\mathcal{D}$ by $\mu$.

We focus on the following testing problem: Given a mixture $P = \frac{1}{k} \sum_{j=1}^{k} \mathcal{D}_{\mu_j}$ of $k$ distributions from a known location family with $\Delta$-separated locations $\mu_1, \ldots, \mu_k$, we need to determine whether the parameter of some component is close to a given $\mu^* \in \mathbb{R}^d$. We prove the following theorem:

**Theorem F.1.** *Let $P = \frac{1}{k} \sum_{j=1}^{k} \mathcal{D}_\mu$ be a uniform mixture of $k \geq 2$ distributions from the location family defined by $\mathcal{D}$ with $\Delta$-separated locations in $\mathbb{R}^d$. Let $\epsilon < \min\{\Delta/32, \Delta/(32\sqrt{\min\{d, \ln k\}})\}$ and $\mu^* \in \mathbb{R}^d$. There is an algorithm that, given distribution $\mathcal{D}$ and samples from $P$, either "accepts" or "rejects", such that it:*

- *Accepts with probability $\geq 2/3$ if $\min_{j \in [k]} \|\mu_j - \mu^*\|_2 \leq \epsilon/2$.*

- *Rejects with probability $\geq 2/3$ if $\min_{j \in [k]} \|\mu_j - \mu^*\|_2 \geq \epsilon$.*

*The runtime (and thus the sample complexity) of the algorithm is upper bounded by*

$$O\left(\frac{k^2 (\Delta/\epsilon)^4}{\min_{\|\xi\|_2 \leq M} \left|\mathbb{E}_{X \sim \mathcal{D}}\left[e^{i\xi^\top X}\right]\right|^2}\right),$$

*where $M \lesssim \frac{1}{\Delta}(\sqrt{d \log k} + \sqrt{(d + \log k) \log \frac{\Delta}{\epsilon}} + \log \frac{\Delta}{\epsilon})$.*

We remark that Theorem F.1 together with our proof strategy of Theorem 2.1 easily gives the following guarantee for learning the parameters $\mu_1, \mu_2, \ldots, \mu_k$.

**Corollary F.2.** *Under the setting of Theorem F.1, there is an algorithm that, given distribution $\mathcal{D}$ and samples from $P$, outputs $\hat{\mu}_1, \ldots, \hat{\mu}_k$ that are w.h.p. $\epsilon$-close to the actual parameters $\mu_1, \ldots, \mu_k$. The runtime of the algorithm is upper bounded by*

$$O\left(\frac{(\Delta/\epsilon)^4 \cdot (k^3/\delta) \log(k/\delta)}{\min_{\|\xi\|_2 \leq M} \left|\mathbb{E}_{X \sim \mathcal{D}}\left[e^{i\xi^\top X}\right]\right|^2}\right),$$

*where $\delta = \Pr_{X \sim \mathcal{D}}\left[\|X\|_2 \leq \epsilon/2\right]$ and $M \lesssim \frac{1}{\Delta}\left(\sqrt{d \log k} + \sqrt{(d + \log k) \log \frac{\Delta}{\epsilon}} + \log \frac{\Delta}{\epsilon}\right)$.*

We will generalize the techniques that underlie Theorem 2.3, which focuses on the special case where $\mathcal{D}$ is a spherical Gaussian.

## F.1 Proof of Theorem F.1

As in the proof of Theorem 2.3, we first examine the Fourier transform of the mixture $P$. For any $\xi \in \mathbb{R}^d$, we have

$$\mathop{\mathbb{E}}_{X \sim P} \left[ e^{i\xi^\top X} \right] = \frac{1}{k} \sum_{j=1}^{k} \mathop{\mathbb{E}}_{X \sim \mathcal{D}} \left[ e^{i\xi^\top (X+\mu_j)} \right] = \frac{(\mathcal{F}\mathcal{D})(\xi)}{k} A_\mu(\xi),$$

where we define $A_\mu(\xi) := \sum_{j=1}^{k} e^{i\mu_j^\top \xi}$ and shorthand $(\mathcal{F}\mathcal{D})(\xi) = \mathbb{E}_{X \sim \mathcal{D}} \left[ e^{i\xi^\top X} \right]$.

The above identity allows us to estimate $A_\mu(\xi)$ accurately, as long as the magnitude of the Fourier transform is not too small at frequency $\xi$. Note that, here, the definition of $A_\mu(\xi)$ is slightly different from that in Lemma 3.1, which has an extra factor of $e^{-\|\mu_j\|_2^2/4}$. This is because we directly look at the Fourier transform of $P$ without the extra "Gaussian truncation".

As in the Gaussian case, we focus on the expectation of $A_\mu(\xi)$ when $\xi$ is drawn from a truncated Gaussian distribution. We have the following analogue of Lemma 3.2:

**Lemma F.3.** *For any $M, \sigma > 0$ that satisfy $M^2/\sigma^2 \geq 5d$,*

$$T_\mu := \mathop{\mathbb{E}}_{\xi \sim \mathcal{N}(0, \sigma^2 I_d)} [A_\mu(\xi) \cdot \mathbb{1}[\|\xi\|_2 \leq M]] = \sum_{j=1}^{k} e^{-\sigma^2 \|\mu_j\|_2^2/2} + k \cdot O\left( e^{-M^2/(5\sigma^2)} \right),$$

*where the $O(x)$ notation hides a complex number with modulus $\leq x$.*

Now we are ready to prove Theorem F.1.

*Proof of Theorem F.1.* The identity $\mathbb{E}_{X \sim P} \left[ e^{i\xi^\top X} \right] = \frac{(\mathcal{F}\mathcal{D})(\xi)}{k} A_\mu(\xi)$ gives

$$T_\mu = \mathop{\mathbb{E}}_{\xi \sim \mathcal{N}(0, \sigma^2 I_d)} [A_\mu(\xi) \cdot \mathbb{1}[\|\xi\|_2 \leq M]] = \mathop{\mathbb{E}}_{\substack{\xi \sim \mathcal{N}(0, \sigma^2 I_d) \\ X \sim P}} \left[ \frac{k e^{i\xi^\top X}}{(\mathcal{F}\mathcal{D})(\xi)} \cdot \mathbb{1}[\|\xi\|_2 \leq M] \right].$$

Let $\delta_M := \min_{\|\xi\|_2 \leq M} |(\mathcal{F}\mathcal{D})(\xi)|$. The above together with a Chernoff bound shows that we can estimate $T_\mu$ up to any error $\gamma > 0$ using $O(k^2 \gamma^{-2} \delta_M^{-2})$ samples.

By Lemma F.3,

$$\left| T_\mu - e^{-\sigma^2 \|\mu_1\|_2^2/2} \right| \leq \sum_{j=2}^{k} e^{-\sigma^2 \|\mu_j\|_2^2/2} + k e^{-M^2/(5\sigma^2)},$$

assuming that $M^2/\sigma^2 \geq 5d$. We will pick $\sigma$ and $M$ carefully, so that both terms above are at most $\gamma := \sigma^2 \epsilon^2/32$. Let $\widehat{T_\mu}$ be an estimate of $T_\mu$ with error $\leq \gamma$. In the case that $\|\mu_1\|_2 \leq \epsilon/2$, we get

$$\operatorname{Re} \widehat{T_\mu} \geq \operatorname{Re} T_\mu - \gamma \geq e^{-\sigma^2 \|\mu_1\|_2^2/2} - 3\gamma \geq e^{-\sigma^2 \epsilon^2/8} - 3\gamma.$$

Similarly, we have $\operatorname{Re} \widehat{T_\mu} \leq e^{-\sigma^2 \epsilon^2/2} + 3\gamma$ when $\|\mu_1\|_2 \geq \epsilon$. If we pick $\sigma$ such that $\sigma^2 \epsilon^2 \leq 1$, we have

$$e^{-\sigma^2 \epsilon^2/8} - e^{-\sigma^2 \epsilon^2/2} \geq \frac{1}{4} \sigma^2 \epsilon^2 = 8\gamma.$$

This means that the range of $\operatorname{Re} \widehat{T_\mu}$ when $\|\mu_1\|_2 \leq \epsilon/2$ is disjoint from that when $\|\mu_1\|_2 \geq \epsilon$, which allows the tester to correctly distinguish the two cases.

Finally, it follows from Claim 3.3 and elementary algebra that all the conditions that we need can be satisfied by picking

$$\sigma^2 = \frac{512}{\Delta^2} \left( \min\{d, \ln k\} + \ln \frac{\Delta}{\epsilon} \right) \quad \text{and} \quad M^2 = 10\sigma^2 \left( d + \ln k + \ln \frac{\Delta}{\epsilon} \right).$$

The time complexity of the testing algorithm is then upper bounded by

$$O\left( k^2 \gamma^{-2} \delta_M^{-2} \right) \lesssim \frac{k^2 (\Delta/\epsilon)^4}{\min_{\|\xi\|_2 \leq M} |(\mathcal{F}\mathcal{D})(\xi)|^2},$$

where

$$M = \frac{\sqrt{5120}}{\Delta} \sqrt{\left( \min\{d, \ln k\} + \ln \frac{\Delta}{\epsilon} \right) \left( d + \ln k + \ln \frac{\Delta}{\epsilon} \right)}$$

$$\lesssim \frac{1}{\Delta} \left( \sqrt{d \log k} + \sqrt{(d + \log k) \log \frac{\Delta}{\epsilon}} + \log \frac{\Delta}{\epsilon} \right).$$

$\square$

## F.2 Examples

We give a few concrete applications of Theorem F.1 and Corollary F.2. For simplicity, we focus on the parameter regime that $d = 1$, $\Delta = O(1)$ and $\Delta/\epsilon = O(1)$, where the runtime of the learning algorithm (from Corollary F.2) can be simplified into

$$O\left( \frac{(k^3/\delta) \log(k/\delta)}{\min_{|\xi| \leq M} | \mathbb{E}_{X \sim \mathcal{D}} [e^{i\xi X}]|^2} \right)$$

for $\delta = \Pr_{X \sim \mathcal{D}} [|X| \leq \epsilon/2]$ and some $M = O\left( \frac{\sqrt{\log k}}{\Delta} \right)$. We note that for all of the following applications, it holds that $\delta = \Omega(\epsilon) = \Omega(\Delta)$. Also recall that the $\mathbb{E}_{X \sim \mathcal{D}} [e^{i\xi X}]$ term is simply the characteristic function of $\mathcal{D}$ at $\xi$.

- **Unit-Variance Gaussian.** For $\mathcal{D} = \mathcal{N}(0, 1)$, the characteristic function of $\mathcal{D}$ is given by $e^{-\xi^2/2}$. This implies

$$\min_{|\xi| \leq M} \left| \mathbb{E}_{X \sim \mathcal{D}} [e^{i\xi X}] \right|^2 = e^{-M^2},$$

  and the runtime reduces to $k^{O(1/\Delta^2)}$. Note that Theorem 2.3 gives a runtime of $O(k^2) \cdot e^{O(\min\{\log k, \log(1/\Delta)\}/\Delta^2)}$ for the one-dimensional unit-variance Gaussian case, which is strictly better than $k^{O(1/\Delta^2)}$ when $1/\Delta$ is sub-polynomial.

- **Cauchy distribution.** The Cauchy distribution with location parameter 0 and scale parameter 1 has probability density function $f(x) = \frac{1}{\pi(1+x^2)}$ and characteristic function $\xi \mapsto e^{-|\xi|}$. Thus, we can learn a mixture of $k$ Cauchy distributions (with unit scale) in time

$$O((k^3/\Delta) \log(k/\Delta)) \cdot e^{O(\sqrt{\log k}/\Delta)} = O(k^3) \cdot e^{O(\sqrt{\log k}/\Delta)},$$

  which is polynomial in $k$ for $\Delta = \Omega(1/\sqrt{\log k})$, and almost cubic in $k$ for $\Delta = \Omega(1)$.

- **Logistic distribution.** The logistic distribution with location 0 and scale 1 has PDF $f(x) = \frac{e^{-x}}{(1+e^{-x})^2}$ and CF $\xi \mapsto \frac{\pi\xi}{\sinh \pi\xi}$. Then, using the inequality $\left| \frac{\sinh(x)}{x} \right| \leq e^{|x|}$, we obtain a learning algorithm with the same runtime as in the Cauchy case: $O(k^3) \cdot e^{O(\sqrt{\log k}/\Delta)}$.

- **Laplace distribution.** The Laplace distribution with fixed scale 1 and location 0 has a characteristic function of $\xi \mapsto \frac{1}{1+\xi^2}$. Applying Corollary F.2 gives a learning algorithm with runtime

$$O\left( (k^3/\Delta)(1 + M^2)^2 \log(k/\Delta) \right) = \tilde{O}\left( k^3/\Delta^5 \right),$$

  which is polynomial in both $k$ and $1/\Delta$.

## F.3 Application #1: Mixture of Exponential Distributions

An important family of distributions over $\mathbb{R}$ that is not included in the location family is the family of exponential distributions, $\{\text{Exp}(\lambda) : \lambda > 0\}$. Nevertheless, we can learning mixtures of exponential distributions using Corollary F.2 and a simple reduction: When $X \sim \text{Exp}(\lambda)$, the random variable $Y := -\ln X$ has a density of $f(x) = \lambda e^{-x} \cdot e^{-\lambda e^{-x}}$, which is exactly the Gumbel distribution with location $\ln \lambda$ and scale 1. The problem is then reduced to learning a mixture of unit-scale Gumbel distributions.

The characteristic function of a Gumbel distribution with location $0$ and scale $1$ is given by

$$\xi \mapsto \Gamma(1 - i\xi),$$

where the Gamma function has modulus

$$|\Gamma(1 - i\xi)| = \sqrt{\frac{\pi\xi}{\sinh(\pi\xi)}} \geq e^{-(\pi/2)|\xi|}.$$

Applying Corollary F.2 then shows that, assuming that $\ln\lambda_1, \ldots, \ln\lambda_k$ are $\Delta$-separated, we can recover these $k$ parameters up to error $O(\Delta)$ in time $O(k^3) \cdot e^{O(\sqrt{\log k}/\Delta)}$. Equivalently, we can recover the parameters $\lambda_1, \ldots, \lambda_k$ up to a multiplicative factor of $e^{O(\Delta)}$ and a permutation.

## F.4 Application #2: Mixture of Linear Regressions in One Dimension

In one dimension, a mixture of $k$ linear regressions is specified by $k$ weights $w_1, \ldots, w_k \in \mathbb{R}$. A labeled data point $(X, Y)$ from the model is sampled by drawing $X \sim \mathcal{N}(0, 1)$, $j \sim \text{Uniform}([k])$ and drawing $Y \sim \mathcal{N}(w_j x, 1)$.

There is a simple reduction from this setting to learning mixtures of unit-variance Gaussians. Note that conditioning on the realization of $X$ and $j$, $Y/X$ is distributed as $\mathcal{N}(w_j, 1/X^2)$. Thus, if $|X| \geq 1$ and we draw $\delta \sim \mathcal{N}(0, 1 - 1/X^2)$, $Y/X + \delta$ follows a Gaussian distribution with mean $w_j$ and variance $1$. Since $|X| \geq 1$ happens with probability $\Omega(1)$, we reduce the problem of learning $w_1, \ldots, w_k$ to learning the means of a mixture of $k$ unit-variance Gaussians, with a constant factor blowup in the time and sample complexities. Therefore, applying Theorem 1.1 shows that if the weights $w_1, \ldots, w_k$ are $\Delta$-separated for $\Delta = \Omega\left(\sqrt{\frac{\log\log k}{\log k}}\right)$, we can recover the weights up to an $O(\Delta)$ error in $\text{poly}(k)$ time.