# OpenReview forum: "A Fourier Approach to Mixture Learning"
_NeurIPS.cc/2022/Conference — NeurIPS 2022 Accept_

### Official Review · Reviewer_6EhS · 2022-07-10

**Rating:** 7
**Confidence:** 2
**Soundness:** 3 good
**Presentation:** 3 good
**Contribution:** 3 good

**Summary:**

The authors consider the problem of estimating means $(\mu_i)_{i=1}^k$  from $m$ sample of the mixture

$\sum_{i = 1}^k \mathcal{N}(\mu_i, I_d)$, up to a small error. The ability to estimate the means primarily depend on the minimum separation between all pairs of means, the number of samples $m$, dimension of samples $d$, and number of clusters $k$. It also desirable to estimate the means, up to an error, with runtime that is polynomial in the parameters. The authors answer this problem in the low dimension regime where $d =o(\log k)$, which has been an open problem. Their main result provides an algorithm that provided samples from the mixture, efficiently learns the parameters in dimension $O(\log k/\log \log k)$ if the minimum separation between the means is at least $\frac{d}{\sqrt{\log k}}$. The runtime is shown to be polynomial in $k$. Their algorithm considers the Fourier transform of the mixture and estimates the corresponding characteristic function by carefully designing a sampling strategy for the frequency in this Fourier domain. I think the paper is well-written and clear.

**Questions:**

- Is there a quantifiable (in parameter like $d$, $\Delta$, $k$, $\dots$) number of sample lower bound required for estimating the means to a fixed error?
- Echoing the points above, what are the practical limitations of the algorithm?



**Limitations:**

The paper very briefly mentions its limitations.


**Strengths And Weaknesses:**

Strengths:

- The paper is well-written and clear. The authors clearly explain the problem, outlining their contributions. Their proof for $d=1$ provides a good initiation for the general proof.
- The problem being considered is a well-studied problem and, to my knowledge, the contribution seem significant.


Weakness:
- While the paper is primarily of a theoretical nature, the authors provide an algorithm to estimate the means of the spherical Gaussian. If possible, it would be very interesting to see numerical performance of this algorithm, or a discussion on why this is not feasible.

---

> ### Author Response · Authors · 2022-08-01
> **Reply to Reviewer 6EhS**
>
> Thank you for your review and your questions on our submission!
>
> **Question 1.** A quantifiable sample complexity lower bound in $d, \Delta, k$?
>
> **Answer:** A more explicit sample complexity lower bound can be inferred from Theorem E.1 in the appendix. In the $d \le \frac{\ln k}{\ln\ln k}$ regime, the lower bound roughly scales as $e^{\Omega(d^2/\Delta^2)}$ and nearly matches the upper bound in Remark 2.2 (up to a $\log(d/\Delta^2)$ factor in the exponent).
>
> **Question 2.** Numerical performance of this algorithm?
>
> **Answer:** The main purpose of this work is to close the gap on the *asymptotic* bounds on the necessary separation needed for learning GMMs. Therefore, the simplicity of the algorithm and its analysis was prioritized over the empirical performance, as done in the long line of work on GMM learning algorithms with provable guarantees (e.g., [Dasgupta, 1999], [Vempala and Wang, 2004], [Regev and Vijayaraghavan, 2017]) that we build upon.
>
> As a result, the constant factors in the algorithms and the bounds were not optimized. We leave it as an interesting open direction to design a more practical GMM learning algorithm that also achieves a near-optimal asymptotic guarantee.

---

### Official Review · Reviewer_eNbi · 2022-07-11

**Rating:** 6
**Confidence:** 2
**Soundness:** 3 good
**Presentation:** 3 good
**Contribution:** 3 good

**Summary:**

This paper provides a Fourier approach to the problem learning mean parameters of spherical Gaussians and its computation complexity. In the low-dimensional regime, the computation complexity of the algorithm is shown to be polynomial in the number of clusters. The computation complexity bound is examined carefully with respect to the magnitude of the the number of dimensions, the number of clusters, and the separateness of the mean parameters. The algorithm can also be extended to learn mixtures of other well-known local families of distributions.

**Questions:**

1. Can you provide some simulation studies with the proposed algorithms and demonstrate the complexity with some figures? It would be very helpful to understand the theory better (especially to differentiate the regimes in Figure 1) and to show that the algorithm works. With the considered setting, simulation studies should not be very difficult.

2. How can we compare the complexity of this Fourier-based algorithm to well-known algorithm solving mixture models such as Expectation-Maximization? It is worth mentioning this in the Related Work section.

**Ethics Review Area:**

["I don’t know"]

**Limitations:**

The considered problem is of interest to the Mixtures Learning community. A limitation of this paper is the lack of experimental studies.

**Strengths And Weaknesses:**

**Strength**: The writing style of the paper helps the reader to quickly understand its main contributions without going too deeply into details with several technical terms when bounding the computation complexity. The proofs in Appendix are clear and carefully written.

**Weakness**: The paper lacks experiment components. Without some simulation studies, it is hard to understand the advantages of this approach to other algorithms in the literature.

---

> ### Author Response · Authors · 2022-08-01
> **Reply to Reviewer eNbi**
>
> Thank you for your review and your questions on our submission!
>
> **Question 1.** Can you provide some simulation studies?
>
> **Answer:** The main purpose of this work is to close the gap on the *asymptotic* bounds on the necessary separation needed for learning GMMs. Therefore, the simplicity of the algorithm and its analysis was prioritized over the empirical performance, as done in the long line of work on GMM learning algorithms with provable guarantees (e.g., [Dasgupta, 1999], [Vempala and Wang, 2004], [Regev and Vijayaraghavan, 2017]) that we build upon.
>
> As a result, the constant factors in the algorithms and the bounds were not optimized. We leave it as an interesting open direction to design a more practical GMM learning algorithm that also achieves a near-optimal asymptotic guarantee.
>
> **Question 2.** Compare the complexity of the Fourier-based algorithm to well-known algorithms like EM.
>
> **Answer:** To the best of our knowledge, unfortunately there are no corresponding results for EM for the settings considered in the paper. Most provable guarantees of the EM algorithm either address the $k = 2$ case (e.g., [Xu et al., 2016], [Balakrishnan et al., 2017] and [Daskalakis et al., 2017]), or only analyze the *local* convergence (e.g., [Kwon and Caramanis, 2020]). One exception is [Dasgupta and Schulman, 2007] mentioned in related work, which shows that a variant of EM succeeds when $d \gg \log k$ and the separation is $\Delta = \Omega(d^{1/4}\log^{1/4}(dk))$.
>
> [Balakrishnan et al., 2017] Sivaraman Balakrishnan, Martin J. Wainwright, Bin Yu. Statistical guarantees for the EM algorithm: From population to sample-based analysis. The Annals of Statistics, 2017
>
> [Daskalakis et al., 2017] Constantinos Daskalakis, Christos Tzamos, Manolis Zampetakis. Ten steps of EM suffice for mixtures of two Gaussians. COLT 2017
>
> [Kwon and Caramanis, 2020] Jeongyeol Kwon, Constantine Caramanis. The EM Algorithm gives Sample-Optimality for Learning Mixtures of Well-Separated Gaussians. COLT 2020
>
> [Xu et al., 2016] Ji Xu, Daniel J. Hsu, Arian Maleki. Global analysis of expectation maximization for mixtures of two Gaussians. NIPS 2016

---

> > ### Comment · Reviewer_eNbi · 2022-08-05
> > **Post-rebuttal Comment**
> >
> > Hi, thank you for answering my questions. I have understood the literature on the mixture learning complexity better by reading the answers.
> >
> > For the score, I still keep the score 6 because the authors do not provide any code or simulation for the proposed algorithm in the paper. I think the paper will be much stronger and have more audience if there is an experimental section.

---

### Official Review · Reviewer_phfG · 2022-07-17

**Rating:** 7
**Confidence:** 3
**Soundness:** 4 excellent
**Presentation:** 3 good
**Contribution:** 4 excellent

**Summary:**

This paper considers the problem of learning Gaussian mixture models; specifically estimating the means of a uniform spherical mixture of $k$ Gaussians with identity covariance matrix in $d$ dimensions. A common parametrization is the minimum separation ($\Delta$) between any mean-pair. Previous works have shown that for polynomial sample complexity a) if $\delta$ is $o(\sqrt{logk})$ then $d$ cannot be $\Omega(\log k)$, b) $\delta = \Omega(\sqrt{log k })$ is possible, and c) $\delta = o(1/\sqrt{log k })$ is not possible. This paper nails down the $\delta$, $d$ tradeoff in the intermediate $\delta$ region. Specifically it shows that $\delta = d/\sqrt{log k}$ is optimal.

The crux of the algorithm is two fold. First sample ln k/p samples from the mixture where p is the probability that a sample from the mixture is \eps close to some cluster's mean. Then the key part of the algorithm is to test if a given guess of the mean is close to one of the mean vectors or far away from it. This is done by estimating the expectation of the Fourier transform under truncated Gaussian as a statistic.

**Questions:**

Apart from the presentation comment, I don't have other questions.

**Limitations:**

The limitations section in the paper is adequate.

**Strengths And Weaknesses:**

This paper settles a gap in existing literature about a fundamental problem that the machine learning community has studied for a while. The analysis is based on a novel technique that possibly might generalize to other mixture models.

I don't have any major complaints. Nevertheless, I think the presentation of the paper could be vastly improved to make it stronger. For example, section 1.2 appears out of nowhere - it is not apiori clear why one might want to consider a testing problem. Even in the later sections, the flow of the algorithm is not clear which is particularly not ideal because only background forms the first 7 out of 9 pages of the submission. It would be much better if main ideas of the algorithm were in the submission instead of the appendix. For instance, it would be cleaner for the main submission to have more details about why drawing samples from the mixture is sufficient to give good mean guesses.

---

> ### Author Response · Authors · 2022-08-01
> **Reply to Reviewer phfG**
>
> Thank you for the suggestions on the presentation of the paper! We will add the following clarifying paragraphs to motivate the testing version, and to explain the reduction from learning to testing. We will also use the additional content page to move the proof of Theorem 2.1 from Appendix C to the main paper.
>
> **Question 1.** Why do we want to consider the testing problem?
>
> **Answer: (Add to Line 99)** Note that this testing problem is not harder than estimating $\mu_1, \ldots, \mu_k$ up to error $\epsilon / 3$---in the former case that $\mu_1 = 0$, one of the mean estimates would fall into $[-\epsilon/3, \epsilon/3]$, whereas all of them must be outside $(-2\epsilon/3, 2\epsilon/3)$ in the latter case. Conversely, as we will prove in Section 2, an algorithm for the testing version can be used for recovering the means as well.
>
> **Question 2.** Why is drawing samples from the mixture sufficient for generating good mean guesses?
>
> **Answer: (Add to Line 281)** Note that if a sample $X_i$ comes from the $j$-th component $\mathcal{N}(\mu_j, I_d)$, there is a decent probability (e.g., $\Omega(\epsilon)$ when $d = 1$) that $X_i$ is $\epsilon$-close to $\mu_j$. Thus, we can guarantee that $X_1, \ldots, X_N$ contain good guesses for all the $k$ means for moderately large $N$.

---

> > ### Comment · Reviewer_phfG · 2022-08-08
> > **Post-Rebuttal comment**
> >
> > Thank you for your response. I am satisfied with it and have no further clarifications.

---

### Meta-Review · Area_Chair_hDKd · 2022-08-24

**Recommendation:** Accept
**Confidence:** Less certain

**Metareview:**

Overall the reviewers felt that this paper should be accepted because it provides a nice theoretical result that pins down the sample vs. gap tradeoff for learning mixtures of Gaussians in low-dimension -- a well studied and interesting problem.

**Award:**

No

---

### Decision · Program_Chairs · 2022-09-14

Accept